# RETHINKING LLM EVALUATION: CAN WE EVALUATE LLMS WITH 200× LESS DATA?

**Shaobo Wang**[*†1,2]   **Cong Wang**[*1]   **Wenjie Fu**[*1,3]   **Yue Min**[1,4]   **Mingquan Feng**[1]   **Isabel Guan**[4]
**Xuming Hu**[4]   **Conghui He**[5]   **Cunxiang Wang**[6]   **Kexin Yang**[2]   **Xingzhang Ren**[2]   **Fei Huang**[2]
**Dayiheng Liu**[2]   **Linfeng Zhang**[✉1]

[1]Shanghai Jiao Tong University   [2]Qwen Team, Alibaba Group   [3]Fudan University

[4]The Hong Kong University of Science and Technology   [5]Shanghai AI Lab   [6]ZhipuAI

 Code: https://github.com/gszfwsb/EssenceBench

## ABSTRACT

Benchmark suites for large language models are growing faster than our ability to pay for them. Even when training is already expensive, many use cases require repeated evaluation across many checkpoints, variants, and competing systems, and the steady expansion of benchmark suites increasingly turns evaluation into a bottleneck in tokens and compute. This scale changes what "useful data" means. Instead of asking whether an instance is good for training one model, we ask *which instances are necessary to keep the collective ordering of many models stable.* We analyze redundancy at the instance level and find repetition in both the text and the ranking patterns induced across models. Based on this observation, we formulate benchmark compression as a subset optimization problem that targets accurate score reconstruction and ranking preservation at the same time. We propose EssenceBench, a coarse-to-fine framework with three stages: redundancy-aware filtering with text and ranking signals, fitness-driven subset search with an iterative genetic algorithm and a fixed surrogate predictor, and attribution-guided refinement for better coverage under tight budgets. Across multiple leaderboards, EssenceBench achieves lower reconstruction error and stronger ranking preservation than prior approaches while reducing selection time. On HellaSwag with 10K instances, EssenceBench preserves 95% of model rankings within a 5% shift using only 50 instances, a 200× compression.

## 1 INTRODUCTION

Large language models (LLMs) have advanced rapidly, exemplified by GPT-4 (Achiam et al., 2023). Benchmark design has expanded accordingly, moving from single-task NLP tests to broad suites spanning multilingual understanding (Zhao et al., 2024), long-context reasoning (Kuratov et al., 2024), instruction following (Yin et al., 2023), mathematical reasoning (Shao et al., 2024), context learning (Dou et al., 2026), code generation (Nam et al., 2024), multidisciplinary knowledge (Zhang et al., 2025), and tool use (Chen et al., 2024). Aggregation platforms such as OpenCompass (Contributors, 2023) make these suites easy to run at scale, but they also make evaluation a recurring expense. In practice, evaluation is repeated across checkpoints, ablations, and competing systems, and steadily growing benchmark suites become a growing burden in tokens and compute.

This paper asks how to reduce benchmark size without changing what a leaderboard would conclude. When a benchmark is used to compare many models, as in the Open LLM Leaderboard[1] (Fourrier et al., 2024), the main output is often the ordering of models. *Benchmark compression should therefore answer a different question: which instances are necessary to keep the collective ordering stable.* The goal is to keep the induced ranking structure intact while reducing evaluation cost.

---

* Equal contribution.   ✉ Corresponding author.   † Project Leader.
[1]https://huggingface.co/datasets/open-llm-leaderboard-old/results

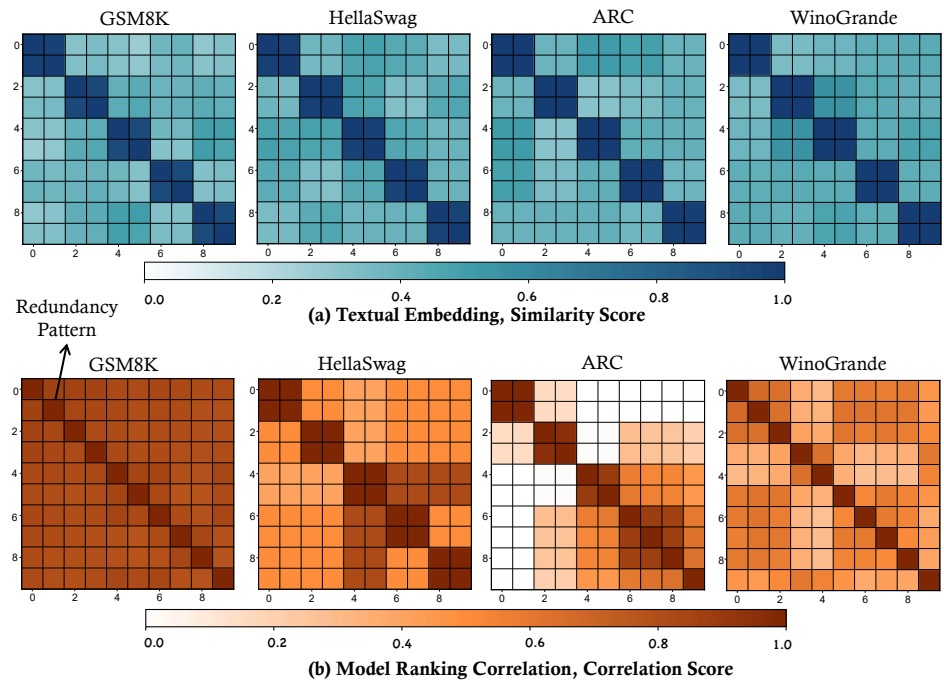

Figure 1: Prevalent redundancy across widely used benchmark datasets. Based on 10 randomly sampled instances per dataset, panel (a) depicts the *text embedding similarity* (Definition 3), reflecting semantic overlap among instances, and panel (b) presents the *ranking embedding similarity* (Definition 4), measured through consistency of model performance rankings across sampled subsets.

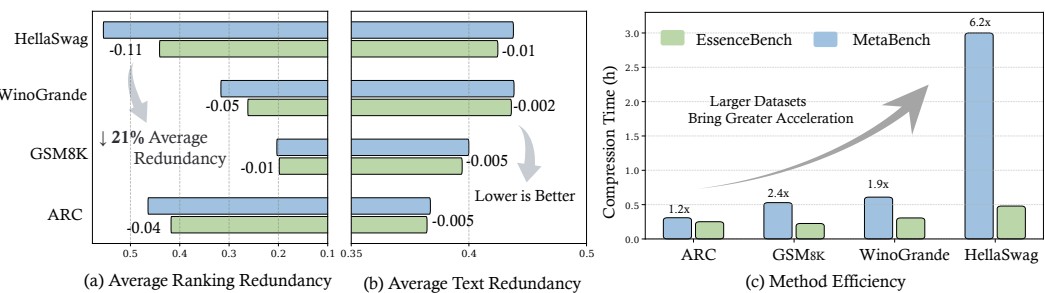

Figure 2: Comparison of existing benchmark compression approaches and our EssenceBench. (a) ranking and (b) text redundancy comparison and (c) compression time comparison.

We view benchmark compression as a subset selection problem with two coupled goals: reconstructing full-benchmark scores and preserving the model ranking. The main challenge is interaction among instances. Redundancy arises not only in the surface text, where prompts are semantically similar, but also in model behavior, where different items induce nearly the same ordering across models. These observations call for objectives and selection signals that reflect both instance-level content and leaderboard-level ordering. Concretely, we quantify text-level redundancy (Definition 3) and ranking-level redundancy (Definition 4) to filter low-value items, and then search for a compact subset that preserves the overall ordering under a fixed budget.

To address this problem, we propose EssenceBench, a coarse-to-fine compression framework with three stages. We first apply a redundancy-aware filter that uses text and ranking signals to shrink the search space. We then run an iterative genetic algorithm to search for compact subsets, scoring each candidate with a fixed surrogate model that reconstructs full-benchmark performance from subset accuracy. Finally, we estimate per-instance attributions from elite subsets and use this signal to guide group-wise refinement, improving coverage and avoiding local optima. Figure 3 provides an overview of the framework, and Figure 2 highlights the empirical gains in both fidelity and efficiency. Our contributions are listed as follows:

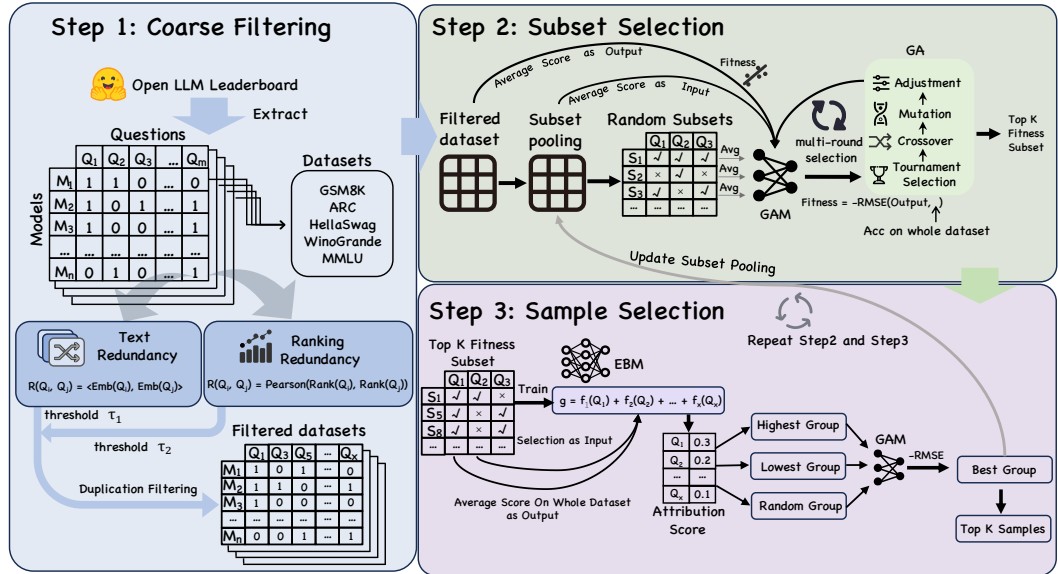

Figure 3: The pipeline of EssenceBench. (I) **Coarse Filtering**. By extracting the binary score matrix for each benchmark and computing both text-level and ranking-level redundancies, samples that exceed thresholds are removed. (II) **Subset Selection**. A genetic algorithm (GA) searches over subsets. Fitness is evaluated by the error of predicted performance, and subsets are optimized via tournament selection, crossover, mutation, and adjustment. (III) **Sample Selection**. Sample attributions are estimated from top-performing subsets and used to build candidate groups. GA is then reapplied within each group to identify the most representative and informative subset.

- We define two complementary notions of redundancy for benchmark compression: **text redundancy** (Definition 3) and **ranking redundancy** (Definition 4), capturing semantic overlap and similarity in induced model-ordering patterns, respectively.

- Based on these definitions, we conduct a sample-level analysis and find that modern benchmarks exhibit substantial redundancy in both text and ranking behavior, suggesting that many instances contribute limited marginal information for distinguishing models.

- We propose EssenceBench, a coarse-to-fine benchmark compression framework that combines redundancy-aware filtering, iterative GA-based subset search with a fixed surrogate, and attribution-guided refinement for robust subset selection under tight budgets.

- Empirically, across multiple leaderboards, EssenceBench achieves lower reconstruction error and stronger ranking preservation than prior approaches. On HellaSwag with 10K instances, it keeps 95% of model rankings within a 5% shift using only 50 instances, achieving $200\times$ compression. On GSM8K with a subset size of 500, EssenceBench reduces RMSE by 60.7% relative to MetaBench. Moreover, on GSM8K and WinoGrande, EssenceBench already surpasses MetaBench with only 200 examples, showing higher data efficiency under tight budgets.

## 2 RELATED WORKS

**Large Language Model Evaluation.** Benchmarks are indispensable for measuring LLM capabilities and catalyzing research. Standard NLU and reading comprehension tasks include GLUE (Wang et al., 2018) and SQuAD (Rajpurkar et al., 2016); mathematical reasoning is tested by GSM8K (Cobbe et al., 2021), MATH (Hendrycks et al., 2021b), GSM-Plus (Li et al., 2024a), and MathQA (Amini et al., 2019); multimodal mathematical reasoning by MathVista (Lu et al., 2023); advanced STEM and graduate-level question answering by GPQA (Rein et al., 2024); complex reasoning by BBH (Suzgun et al., 2023) and MuSR (Sprague et al., 2023); coding ability via HumanEval (Chen et al., 2021) and MBPP (Austin et al., 2021); instruction following by IFEval (Zhou et al., 2023b); agentic tool use by LiveMCPBench (Mo et al., 2025); and multiple-choice and commonsense reasoning by MMLU (Hendrycks et al., 2021a), ARC-Challenge (Clark et al., 2018), Winogrande (Sakaguchi et al., 2021) and HellaSwag (Zellers et al., 2019). Platforms such as Open LLM Leaderboard (Fourrier et al.,

2024) and OpenCompass (Contributors, 2023) provide unified pipelines and live leaderboards, but they rarely consider evaluation cost and do not address how to reduce benchmarks without changing the leaderboard conclusions.

**Benchmark Compression for Evaluation.** Benchmark compression operates on evaluation sets and aims to reduce test-time cost while preserving reliable comparisons across models (Li et al., 2024b). Representative methods include TinyBenchmark (Polo et al., 2024), MetaBench (Kipnis et al., 2025), Reliable evaluation based on amortized difficulty estimation (Truong et al., 2025), and SMART filtering (Gupta et al., 2025). While these methods improve efficiency, many do not explicitly model instance interactions induced by redundancy in both text and ranking behavior, and scalable subset search under strict budgets remains challenging.

**Data Selection for Training.** Data selection for training focuses on constructing smaller or cleaner pretraining and finetuning corpora. Common approaches include large-scale filtering to remove low-quality, toxic, or duplicated samples (Raffel et al., 2020; Marion et al., 2023; Zhang et al., 2022; Abbas et al., 2023), as well as subset selection strategies that aim to retain downstream performance with fewer examples, often using heuristics or IRT-based scoring (Zhou et al., 2023a; Wang et al., 2023; Ivison et al., 2023). More recent studies explore using LLMs themselves to score or rank data (Xia et al., 2024; Liu et al., 2023). However, these methods are primarily optimized for learning efficiency and typically do not target benchmark-level score reconstruction or ranking consistency.

## 3 METHODOLOGY

This section describes how we compress a benchmark into a small subset that remains predictive for *many* models. The core challenge is that evaluation redundancy appears at two levels: prompts can be semantically similar, and different prompts can still induce nearly identical cross-model behavior. Our solution, EssenceBench, follows a coarse-to-fine storyline: we first remove near-duplicate instances (cheap pruning), then search for a score-reconstructive coreset (global optimization), and finally refine it for better coverage under tight budgets (local correction), as illustrated in Figure 3.

### 3.1 PRELIMINARIES: BENCHMARK COMPRESSION

Benchmark compression can be framed as selecting a $k$-element subset (a coreset) from a benchmark of size $N$ under a strict evaluation budget. In the leaderboard setting, the subset should remain predictive for *many* models: it should accurately reconstruct benchmark scores and, crucially, preserve the induced *relative ordering* among models. Searching over all $k$-subsets is a classic NP-hard combinatorial optimization problem. We formalize benchmark compression as follows.

**Definition 1** (Benchmark Compression). *Let $\mathcal{D} = \{x_1, \ldots, x_N\}$ be a benchmark dataset, and let $g : 2^{\mathcal{D}} \to \mathbb{R}$ be an aggregate scoring function that assigns a performance score to any subset of $\mathcal{D}$. Given a budget $k < N$, the compression problem seeks a $k$-element subset $\tilde{\mathcal{D}} \subseteq \mathcal{D}$ that best reconstructs the full-dataset score:*

$$\tilde{\mathcal{D}}^* = \underset{\tilde{\mathcal{D}} \subseteq \mathcal{D}, |\tilde{\mathcal{D}}| = k}{\arg\min} \mathcal{L}\big(g(\mathcal{D}), g(\tilde{\mathcal{D}})\big), \tag{1}$$

*where $\mathcal{L}(\cdot, \cdot)$ represents a suitable error measure.*

Directly optimizing Definition 1 is intractable due to the exponential search space. We therefore leverage public leaderboards (Fourrier et al., 2024), which provide per-instance correctness signals for multiple models. By organizing these into a *score matrix*, benchmark compression reduces to selecting informative columns whose aggregate statistics reconstruct the full-benchmark evaluation.

Let $N_{\text{model}}$ denote the number of evaluated models and let $N$ denote the number of benchmark instances. We denote the *score matrix* by $\mathbf{S} \in \{0, 1\}^{N_{\text{model}} \times N}$, where $S_{i,j}$ is the binary outcome of model $i$ on instance $j$ (1 for correct and 0 otherwise). Let $\mathbf{y} \in \mathbb{R}^{N_{\text{model}}}$ be the full-benchmark accuracy vector, with $y_i = \frac{1}{N} \sum_{j=1}^{N} S_{i,j}$. The resulting concrete formulation is given below.

**Definition 2** (Concrete Formulation of Benchmark Compression). *Let a binary mask represent the selection of a $k$-element subset: $\mathbf{m} \in \{0, 1\}^N$ s.t. $\sum_{j=1}^{N} m_j = k$. By indexing the mask, we obtain the corresponding submatrix of $\mathbf{S}$, denoted by $\mathbf{S_m}$. The aggregate scoring function in Definition 1*

*can then be expressed as a function of the selected columns, i.e., $g(\tilde{\mathcal{D}}) = g(\mathbf{S_m})$. The compression objective becomes:*

$$\min_{\mathbf{m} \in \{0,1\}^N} \mathcal{L}(\mathbf{y}, g(\mathbf{S_m})) \quad \text{s.t.} \sum_{j=1}^{N} m_j = k. \tag{2}$$

## 3.2 A Closer Look at the Sample Redundancy Phenomenon in LLM Benchmarks

As illustrated in Figure 1, many LLM benchmarks exhibit high overlap both in the text of their prompts and in the models' performance rankings in our quantification of sample redundancy, *i.e.* text redundancy (Definition 3) and ranking redundancy (Definition 4). Intuitively, when two examples share similar wording or contain nearly identical behavior across some models, retaining both adds little new information while doubling evaluation costs.

**Definition 3** (Sample Redundancy from Text Perspective). *Let $\mathcal{D} = \{x_1, x_2, \ldots, x_N\}$ be a benchmark dataset where each $x_i$ denotes the textual input. Let $\text{Emb} : x \mapsto \mathbb{R}^{d_{\text{emb}}}$ denote an embedding mapping of the input. The redundancy of a specific sample pair is defined as:*

$$\mathcal{R}_{\text{text}}(i, j) = \langle \text{Emb}(x_i), \text{Emb}(x_j) \rangle. \tag{3}$$

*The redundancy of a sample is then defined as:*

$$\mathcal{R}_{\text{text}}(i) = \frac{\sum_{j \neq i} \langle \text{Emb}(x_i), \text{Emb}(x_j) \rangle}{N - 1}. \tag{4}$$

*The overall redundancy of the benchmark is defined as the average redundancy across all samples:*

$$\mathcal{R}_{\text{text}}(\mathcal{D}) = \frac{\sum_{i=1}^{N} \mathcal{R}_{\text{text}}(i)}{N}. \tag{5}$$

**Definition 4** (Sample Redundancy from Ranking Perspective). *Let $\mathcal{D} = \{x_1, x_2, \ldots, x_N\}$ denote the benchmark dataset containing $N$ instances, and let $N_{\text{model}}$ denote the number of evaluated models. For each instance $x_j$, we define its* performance vector *across models as $\mathbf{s}_j \in \mathbb{R}^{N_{\text{model}}}$, where the $i$-th element is the outcome (or score) of model $i$ on $x_j$. The ranking redundancy between two instances is quantified by the Pearson correlation between their performance vectors:*

$$\mathcal{R}_{\text{rank}}(i, j) = \rho(\mathbf{s}_i, \mathbf{s}_j) = \frac{\text{Cov}(\mathbf{s}_i, \mathbf{s}_j)}{\sigma_{\mathbf{s}_i} \sigma_{\mathbf{s}_j}}, \tag{6}$$

*Here $\rho \in [-1, 1]$. A high correlation indicates that models exhibit highly similar behavior on both instances, suggesting redundant signal for distinguishing models.*

These two definitions are intuitively derived from both human and LLM perspectives. Specifically, textual redundancy quantifies semantic similarity, while ranking redundancy assesses behavioral similarity. When combined, they reveal complementary overlaps that each alone would overlook, facilitating more systematic redundancy elimination and benchmark pruning.

## 3.3 EssenceBench

**Overview.** EssenceBench compresses a benchmark in three steps under a budget $k$. Step 1 removes near-duplicate instances using text and ranking redundancy signals. Step 2 searches for a $k$-instance subset that best reconstructs full-benchmark scores using a genetic algorithm with a fixed surrogate fitness. Step 3 refines the subset using attribution-guided grouping to improve coverage and reduce local-optimum bias. Figure 3 summarizes the pipeline.

**Step 1: Coarse filtering (remove near-duplicates).** We prune the benchmark using two complementary redundancy signals: *text redundancy* (Definition 3) and *ranking redundancy* (Definition 4). Given thresholds $\tau_{\text{text}}$ and $\tau_{\text{ranking}}$, we scan instances in their original order and keep the first occurrence among any highly redundant pair. Concretely, instance $x_i$ is kept if it is not overly similar to any earlier instance under *both* signals. Let $\epsilon_i \in \{0, 1\}$ denote whether $x_i$ is kept:

$$\epsilon_i = \prod_{j=1}^{i-1} \mathbf{1}\big(\mathcal{R}_{\text{text}}(j, i) \leq \tau_{\text{text}} \ \wedge \ \mathcal{R}_{\text{ranking}}(j, i) \leq \tau_{\text{ranking}}\big), \tag{7}$$

where $\mathbf{1}(\cdot)$ denotes the indicator function. Therefore, the filtered benchmark is $\mathcal{D}_{\text{filtered}} = \{ x_i \mid \epsilon_i = 1 \}$ with size $M$ (we reuse $M$ to denote the number of remaining instances after filtering). This step cheaply removes the most obvious duplication, so later searches focus on more latent redundancy.

**Step 2: Fitness-based subset selection (genetic algorithm; GA).** To shrink the filtered benchmark while preserving its predictive power, we search for a $k$-instance subset using a genetic algorithm (GA) (Lambora et al., 2019; Mirjalili, 2018; Mathew, 2012; Mitchell, 1998). A genetic algorithm is a population-based evolutionary search method that iteratively refines candidate solutions through selection and randomized variation. Concretely, GA maintains a set of candidate subsets (a population), scores each candidate by a fitness function, and improves candidates by recombining and perturbing high-fitness ones.

**What happens inside one GA generation.** Each generation consists of four intuitive operations: (i) *evaluate* each candidate subset with the fitness function (Equation 9); (ii) *select* two high-fitness parents via tournament selection; (iii) *mix* parents with crossover and *perturb* with mutation to create a child; and (iv) *repair* the child to satisfy the budget constraint $\|\mathbf{m}\|_1 = k$ (flip bits if needed). Repeated over many generations, GA concentrates probability mass on subsets that best reconstruct full-benchmark scores.

• *Individual and population.* An *individual* is a binary mask $\mathbf{m} \in \{0,1\}^M$ with $\sum_{j=1}^{M} m_j = k$, representing a $k$-instance subset. The *population* is a set of individuals: $\mathcal{P} = \{\mathbf{m}^{(1)}, \ldots, \mathbf{m}^{(N_\mathcal{P})}\}$, where $N_\mathcal{P}$ is the number of the individuals in the population. The final top-$N_\mathcal{E}$ best subsets are also denoted as a set: $\mathcal{E} = \{\mathbf{m}^{(1)}, \ldots, \mathbf{m}^{(N_\mathcal{E})}\}$.

• *Fitness evaluation (fixed surrogate).* Fitness corresponds to minimizing the reconstruction loss in Equation 2. Direct evaluation of $g$ for every candidate subset is expensive, so we train a lightweight surrogate once per GA round: a Generalized Additive Model (GAM) (Hastie, 2017) that maps a model's *subset accuracy* to its full-benchmark accuracy. The GAM is then kept *fixed* during the evolutionary search, ensuring both efficiency and evaluation consistency.

Let the score matrix and full-benchmark accuracies on the filtered benchmark be $\mathbf{S}_{\text{filtered}} \in \{0,1\}^{N_{\text{model}} \times M}$ and $\mathbf{y}_{\text{filtered}} \in \mathbb{R}^{N_{\text{model}}}$. For a subset mask $\mathbf{m} \in \{0,1\}^M$ with $\|\mathbf{m}\|_1 = k$, the subset accuracy of model $i$ is

$$s_i(\mathbf{m}) = \frac{1}{k} \sum_{j=1}^{M} \mathbf{S}_{\text{filtered}}[i,j] \cdot m_j . \tag{8}$$

We then predict $\hat{y}_i = g(s_i(\mathbf{m}))$. Using RMSE on a held-out model set $\mathcal{V}$, we define:

$$\text{fitness}(\mathbf{m}) = -\sqrt{\frac{1}{|\mathcal{V}|} \sum_{i \in \mathcal{V}} (\hat{y}_i - y_i)^2} . \tag{9}$$

This objective is stable across candidates because $g$ is fixed within each GA round.

• *Tournament selection.* In this process, we aim to choose an individual from $\mathcal{P}$ as a *parent* according to fitness (Equation 9). Let $\mathbf{m}^{(a)}, \mathbf{m}^{(b)}$ denote the two parents chosen in this process.

• *Crossover.* In this process, the goal is to get a new individual by combining the information of the two parents. Let $\xi_j$ denote a flag which parent the new individual should follow, the *crossover* process generates a new individual $\mathbf{m}^{(c)}$ by:

$$m_j^{(c)} = (m_j^{(a)} \wedge \xi_j) \vee (m_j^{(b)} \wedge \neg\xi_j), \tag{10}$$

where $\xi_j \sim \text{Bernoulli}(0.5), \ j \in [1, M]$.

• *Mutation.* In this process, the aim is to introduce randomness into the new individual $\mathbf{m}^{(c)}$, let $\lambda_j$ be a flag which sample of an individual should mutate:

$$m_j^{(c)} \leftarrow m_j^{(c)} \oplus \lambda_j, \tag{11}$$

where $\lambda_j \sim \text{Bernoulli}\left(\frac{1}{k}\right), \ j \in [1, M]$.

• *Adjustment.* To guarantee $\mathbf{m}^{(c)}$ has $k$-ones, the *adjustment* process is implemented by randomly setting superfluous ones to zeros, vice versa.

**Step 3: Attribution-based Sample Selection**. To maximize reconstruction fidelity while preserving representational diversity during dataset compression, we train an Explainable Boosting Machine (EBM) (Nori et al., 2019) on the elite mask set $\mathcal{E} = \{\mathbf{m}^{(1)}, \ldots, \mathbf{m}^{(N_{\mathcal{E}})}\}$ from Step 2, assigning each sample in $\mathcal{D}_{\text{filtered}} = \{x_1, \ldots, x_M\}$ a data-specific attribution score; these scores are used to stratify samples into groups, upon which a genetic algorithm (GA) performs optimized selection to balance signal strength and coverage. The result is a compressed dataset $\mathcal{D}_{\text{compressed}}$ of size $P < M$, which retains high-impact instances while preserving underrepresented patterns—achieving efficient, robust, and generalizable compression without compromising reconstruction performance.

To quantify how much each sample contributes to reconstruction accuracy in the predictor model, we first define the *attribution* of each sample within a mask. For a mask $\mathbf{m} \in \mathcal{E}$, define the selected index set $\mathcal{I}(\mathbf{m}) = \{j \,|\, m_j = 1\}$. An EBM $g_{\mathbf{m}}(\mathcal{D}_{\text{filtered}}) = \sum_{j \in \mathcal{F}(\mathbf{m})} f_j^{\mathbf{m}}(x_j)$ learns the training data $\mathcal{T}'$. Let $\mathbf{S}_{\text{filtered},i}$ denote the $i$-th row of $\mathbf{S}_{\text{filtered}}$, then the form of $\mathcal{T}'$ is: $\{\mathbf{S}_{\text{filtered},i}, y_i\}_{i \in \mathcal{T}'}$, which constructs a map from sample score matrix to whole dataset accuracy. The component norm $\|f_l^{\mathbf{m}}\|_2$ in $g_{\mathbf{m}}$ is defined as the *attribution* of sample $l$ in a mask. Aggregating over all attributions of each mask in $\mathcal{E}$ yields the *global attribution* of sample $l$, denoted as $A_l$:

$$A_l = \frac{\sum_{\mathbf{m} \in \mathcal{E}} \mathbf{1}\{j \in \mathcal{I}(\mathbf{m})\} \|f_l^{\mathbf{m}}\|_2}{\sum_{\mathbf{m} \in \mathcal{E}} \mathbf{1}\{j \in \mathcal{I}(\mathbf{m})\}}, \tag{12}$$

where $l \in \mathcal{I}(\mathcal{D}_{\text{filtered}})$. According to the attributions, a tri-partition of the samples can be implemented. With a retention ratio $\alpha \in (0, 1)$, set $q = \lceil \alpha M \rceil = P \ll M$ and create three groups of *equal* size $q$: $G_{\text{high}}, G_{\text{low}}, G_{\text{rand}}$, where $G_{\text{high}}$ contains the $q$ samples with the largest $A_l$, $G_{\text{low}}$ the smallest ones, and $G_{\text{rand}}$ the random ones. The grouping operation deliberately forces subsequent GAs to search in regions of different attributions so that information neglected by the current top-$N_{\mathcal{E}}$ subsets can be rediscovered, while the information that is really significant can be boosted. For every group $G \in \{G_{\text{high}}, G_{\text{low}}, G_{\text{rand}}\}$, GA is used to judge the best group to get $\mathcal{D}_{\text{compressed}}$.

To increase diversity of pruned dataset, we iteratively repeat Step 2 and Step 3, at each round the globally best mask $\mathbf{m}^{\star}$ is updated whenever a lower error is observed. The details are in Appendix A.

# 4 EXPERIMENTS

## 4.1 EXPERIMENTAL SETUP

**Datasets.** We constructed our evaluation using data from the Open LLM Leaderboard (Fourrier et al., 2024). We use 5 standard benchmarks: GSM8K (1K samples), ARC (400 samples), HellaSwag (10K samples), WinoGrande (44K samples), and MMLU (15K samples). To validate the generalization of our approach, we extended experiments to 8 modern benchmarks: LiveMCPBench (agentic tool use), MathVista (multimodal math), GPQA (graduate STEM), BBH (complex reasoning), MATH (advanced math), MUSR (multistep reasoning), IFEval (instruction following), and GSM-Plus (robustness). Data preprocessing followed protocols from MetaBench (Kipnis et al., 2025), utilizing a strict 9:1 train-test split based on stratified sampling of model performance strata.

**Baselines.** We compared EssenceBench against state-of-the-art compression methods MetaBench (Kipnis et al., 2025) and SMART (Gupta et al., 2025). Additionally, we included classic selection strategies: Random Selection, GraNd (Paul et al., 2021), Perplexity (PPL) (Bengio et al., 2003), K-Means Clustering, and MLP-based Selection.

## 4.2 MAIN RESULTS ON STANDARD BENCHMARKS

**Score Reconstruction Performance.**

Tables 1 and 2 report the leaderboard score reconstruction error (RMSE) on 13 benchmarks. Across all datasets and subset sizes, EssenceBench consistently achieves the lowest RMSE. Notably, on GSM8K, EssenceBench with only $k = 200$ samples achieves an RMSE of 0.86, outperforming MetaBench at $k = 500$ (RMSE 0.96), corresponding to a $2.5\times$ improvement in data efficiency. Similarly, on WinoGrande, EssenceBench with $k = 200$ (RMSE 0.78) matches MetaBench with $k = 500$ (RMSE 0.79). Beyond these standard benchmarks, EssenceBench maintains the same trend on more challenging suites (Table 2), e.g., it achieves near-perfect reconstruction on MathVista (RMSE 0.001 at $k = 300$) and remains highly robust on GSM-Plus (RMSE 0.010 at $k = 300$,

Table 1: Prediction error (RMSE ↓) on 5 standard benchmarks using subset size $N \in \{50, 100, \ldots, 500\}$. EssenceBench achieves lower error than all baselines across subset sizes.

| Dataset | Method | Subset Size ($k$) | | | | | | | | | |
|---|---|---|---|---|---|---|---|---|---|---|---|
| | | 50 | 100 | 150 | 200 | 250 | 300 | 350 | 400 | 450 | 500 |
| GSM8K | Random | 3.689 | 2.853 | 2.293 | 1.945 | 1.627 | 1.401 | 1.343 | 1.228 | 1.149 | 1.040 |
| | PPL | 4.194 | 2.699 | 2.301 | 1.922 | 1.719 | 1.515 | 1.359 | 1.300 | 1.199 | 1.130 |
| | GraNd | 3.852 | 2.893 | 2.373 | 2.096 | 1.826 | 1.599 | 1.444 | 1.272 | 1.157 | 1.069 |
| | MetaBench | 3.528 | 2.434 | 2.067 | 1.760 | 1.553 | 1.387 | 1.263 | 1.130 | 1.033 | 0.958 |
| | EssenceBench | **2.769** | **1.667** | **1.152** | **0.864** | **0.718** | **0.610** | **0.559** | **0.504** | **0.456** | **0.377** |
| ARC | Random | 3.153 | 2.446 | 2.165 | 1.731 | 1.450 | 1.319 | 1.153 | 1.098 | 0.934 | 0.902 |
| | PPL | 5.334 | 3.119 | 2.117 | 1.819 | 1.546 | 1.416 | 1.217 | 1.088 | 1.005 | 0.917 |
| | GraNd | 5.334 | 2.947 | 2.148 | 1.840 | 1.602 | 1.350 | 1.157 | 1.044 | 1.000 | 0.943 |
| | MetaBench | 2.741 | 2.077 | 1.684 | 1.447 | 1.248 | 1.110 | 0.977 | 0.949 | 0.863 | 0.749 |
| | EssenceBench | **2.399** | **1.429** | **1.165** | **0.802** | **0.719** | **0.605** | **0.505** | **0.480** | **0.433** | **0.370** |
| HellaSwag | Random | 2.574 | 1.807 | 1.466 | 1.198 | 1.157 | 1.020 | 0.975 | 0.893 | 0.797 | 0.834 |
| | PPL | 2.975 | 1.995 | 1.643 | 1.298 | 1.080 | 0.931 | 0.916 | 0.833 | 0.794 | 0.747 |
| | GraNd | 2.922 | 2.010 | 1.589 | 1.257 | 1.043 | 0.960 | 0.884 | 0.824 | 0.842 | 0.783 |
| | MetaBench | 2.434 | 1.694 | 1.468 | 1.314 | 1.168 | 1.067 | 0.993 | 0.912 | 0.871 | 0.822 |
| | EssenceBench | **2.264** | **1.572** | **1.232** | **1.064** | **0.891** | **0.748** | **0.668** | **0.615** | **0.533** | **0.511** |
| WinoGrande | Random | 3.500 | 2.871 | 2.148 | 1.949 | 1.594 | 1.531 | 1.277 | 1.154 | 1.128 | 0.985 |
| | PPL | 4.269 | 2.748 | 2.340 | 1.935 | 1.791 | 1.775 | 1.571 | 1.471 | 1.399 | 1.318 |
| | GraNd | 4.269 | 2.656 | 2.305 | 2.014 | 1.778 | 1.767 | 1.616 | 1.505 | 1.404 | 1.273 |
| | MetaBench | 2.783 | 2.122 | 1.752 | 1.530 | 1.289 | 1.203 | 1.072 | 0.958 | 0.866 | 0.785 |
| | EssenceBench | **2.509** | **1.399** | **0.979** | **0.777** | **0.631** | **0.558** | **0.510** | **0.452** | **0.413** | **0.391** |
| MMLU | Random | 3.505 | 2.288 | 2.104 | 1.910 | 1.578 | 1.590 | 1.498 | 1.336 | 1.336 | 1.170 |
| | PPL | 8.029 | 9.763 | 10.500 | 8.505 | 8.015 | 7.882 | 7.678 | 7.275 | 7.206 | 6.781 |
| | GraNd | 9.000 | 10.091 | 10.475 | 8.733 | 8.156 | 7.903 | 7.745 | 7.223 | 7.578 | 6.851 |
| | MetaBench | 2.427 | 2.093 | 1.738 | 1.529 | 1.362 | 1.287 | 1.199 | 1.140 | 1.063 | 0.994 |
| | EssenceBench | **2.412** | **1.829** | **1.395** | **1.113** | **1.022** | **0.846** | **0.767** | **0.691** | **0.641** | **0.597** |

Table 2: Prediction error (RMSE ↓) on 8 more challenging benchmarks using subset size $k \in \{50, 150, 300\}$[2]. EssenceBench yields the lowest error across diverse evaluation settings.

| Dataset | Method | $k$=50 | $k$=150 | $k$=300 | Dataset | Method | $k$=50 | $k$=150 | $k$=300 |
|---|---|---|---|---|---|---|---|---|---|
| MathVista (Multimodal) | Random | 8.381 | 8.068 | 8.811 | MATH (Adv. Math) | Random | 3.132 | 1.879 | 1.166 |
| | MetaBench | 2.951 | 2.779 | 4.389 | | MetaBench | 2.389 | 1.472 | 0.990 |
| | Ours | **0.017** | **0.009** | **0.001** | | Ours | **1.895** | **0.668** | **0.475** |
| GPQA (Grad. STEM) | Random | 2.739 | 2.028 | 1.441 | BBH (Reasoning) | Random | 5.022 | 2.383 | 1.842 |
| | MetaBench | 2.117 | 1.660 | 1.207 | | MetaBench | 3.279 | 2.036 | 1.523 |
| | Ours | **1.841** | **0.833** | **0.470** | | Ours | **2.442** | **1.063** | **0.733** |
| IFEval (Instruction) | Random | 4.309 | 2.248 | 1.317 | GSM-Plus (Robustness) | Random | 6.749 | 4.692 | 3.016 |
| | MetaBench | 3.607 | 1.985 | 1.139 | | MetaBench | 0.575 | 0.375 | 0.195 |
| | Ours | **3.603** | **1.239** | **0.555** | | Ours | **0.171** | **0.030** | **0.010** |
| MUSR (Multistep) | Random | 3.100 | 2.102 | 1.414 | LiveMCP (Agentic) | Random | 10.23 | 10.61 | 10.87 |
| | MetaBench | 2.927 | 1.697 | 1.106 | | MetaBench | 8.242 | 8.165 | 8.210 |
| | Ours | **2.538** | **0.949** | **0.412** | | Ours | **0.015** | **0.003** | **0.001** |

vs. MetaBench 0.195). These results demonstrate that our strategy effectively identifies the most informative samples for accurate leaderboard reconstruction.

**Ranking Fidelity.** Beyond RMSE, we evaluated ranking preservation using a suite of metrics (Pearson, Kendall, Rank Stability, Top-50 Acc). As shown in Table 3 (and other tables in Appendix B), EssenceBench consistently achieves higher ranking correlations. For instance, on GSM8K with only 200 samples, our method attains a Kendall correlation of 0.968, surpassing MetaBench's performance

---

[2]LiveMCPBench contains only 93 instances, therefore its three columns correspond to $k \in \{50, 70, 90\}$.

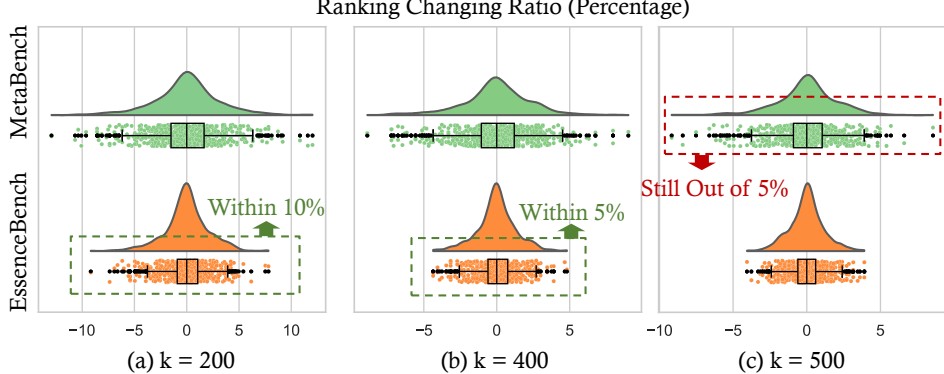

Figure 4: Comparison of ranking change distributions between MetaBench and EssenceBench on the HellaSwag dataset. Our method shows significantly tighter error bounds.

Table 3: Ranking Fidelity on GSM8K. EssenceBench achieves superior correlation by fewer samples.

| Metric | Method | 50 | 100 | 150 | 200 | 250 | 300 | 350 | 400 | 450 | 500 |
|---|---|---|---|---|---|---|---|---|---|---|---|
| Pearson ↑ | MetaBench | 0.991 | 0.995 | 0.997 | 0.998 | 0.998 | 0.999 | 0.999 | 0.999 | 0.999 | 0.999 |
| | EssenceBench | **0.994** | **0.998** | **0.999** | **0.999** | **1.000** | **1.000** | **1.000** | **1.000** | **1.000** | **1.000** |
| Kendall ↑ | MetaBench | 0.888 | 0.912 | 0.934 | 0.937 | 0.949 | 0.950 | 0.956 | 0.960 | 0.964 | 0.967 |
| | EssenceBench | **0.905** | **0.943** | **0.955** | **0.968** | **0.973** | **0.976** | **0.978** | **0.980** | **0.982** | **0.985** |
| Rank Err (within 10%) ↑ | MetaBench | 0.953 | 0.959 | 0.987 | 0.984 | 0.997 | 0.993 | 0.998 | 0.998 | **1.000** | **1.000** |
| | EssenceBench | **0.956** | **0.989** | **1.000** | **1.000** | **1.000** | **1.000** | **1.000** | **1.000** | **1.000** | **1.000** |

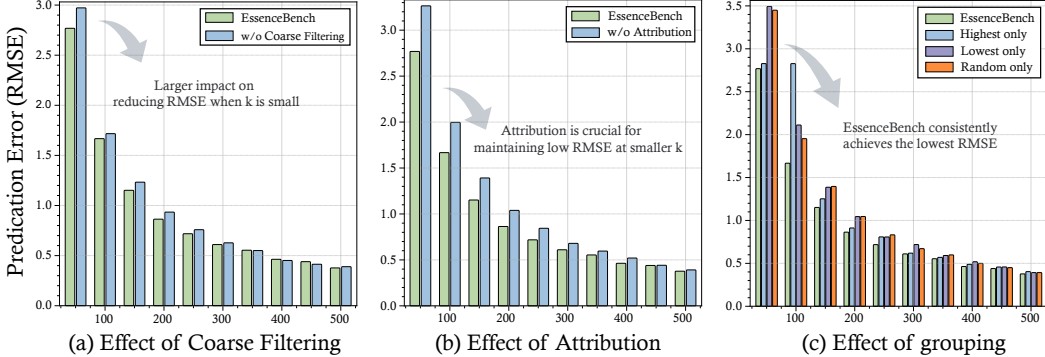

Figure 5: Ablation results on GSM8K, evaluating the effect of (a) coarse filtering, (b) attribution-based selection, and (c) grouping strategies.

at 450 samples (0.964). Figure 4 visually confirms this: with $k = 200$, EssenceBench constrains ranking shifts strictly within 10%, whereas MetaBench exhibits significantly larger deviations.

**Ablation Study.** We analyzed the impact of Coarse Filtering, Attribution-based selection, and Grouping strategies. As illustrated in Figure 5, results indicate that increasing refinement rounds consistently reduces RMSE (e.g., from 2.77 to 2.47 on GSM8K). Coarse filtering is crucial for small subset sizes, removing redundancy that degrades quality. Attribution-guided grouping effectively outperforms random selection when the data budget is tight ($k < 400$).

**Comparison with SMART and other Optimization Baselines.** We compared our approach against K-means, MLP-based selection, and the concurrent work SMART (Gupta et al., 2025). As shown in Table 4 (Left), EssenceBench achieves 2.4-6.8× lower error than K-means and 1.7-3.5× improvement over MLP on GSM8K. Comparing to SMART on ARC-Challenge (Table 4, Right), we achieve better RMSE (1.104 vs 1.193) while using significantly fewer samples (8.53% vs 39.25% of the dataset). As illustrated in Table 5, using SMART as our coarse filtering step yields marginal gains, suggesting our coarse filtering is already highly effective because it needs less information and computation.

Table 4: Left: Comparison against optimization baselines on GSM8K (RMSE ↓). Right: Comparison against SMART on ARC-Challenge.

| Optimization Baselines (GSM8K) | | | | | vs. SMART (ARC-Challenge) | | |
|---|---|---|---|---|---|---|---|
| Method | $k$=50 | $k$=100 | $k$=200 | $k$=400 | Metric | SMART | Ours |
| K-means | 4.133 | 4.131 | 3.300 | 2.757 | Subset Size ($k$) | 460 (39%) | **100 (8%)** |
| MLP-Select | 4.637 | 3.354 | 2.417 | 1.573 | RMSE ↓ | 1.193 | **1.104** |
| EssenceBench | **2.900** | **1.525** | **0.857** | **0.454** | Pearson ↑ | 0.997 | **0.998** |

Table 5: Comparison of EssenceBench and replacing coarse filtering step with SMART on MMLU (RMSE ↓).

| Subset Size ($k$) | 50 | 100 | 150 | 200 | 250 | 300 |
|---|---|---|---|---|---|---|
| EssenceBench | **2.41** | 1.83 | **1.40** | 1.11 | 1.02 | 0.85 |
| SMART as coarse filtering | 2.59 | **1.72** | 1.41 | **1.10** | **0.98** | **0.82** |

## 4.3 ROBUSTNESS, GENERALIZABILITY, AND EFFICIENCY

**Robustness to Hyperparameters.** We vary key hyperparameters in coarse filtering and optimization, including $(\tau_{txt}, \tau_{rank})$, GA population size $N_p$, elite size $N_\epsilon$, and the duplication filtering strategy. Across GSM8K and ARC, RMSE changes remain within $\pm 0.15$, indicating robust performance without careful tuning. Details are reported in Appendix C, Tables 8–12.

**Generalization to Unseen Models.** We test generalization using a hold-out split of 600 unseen leaderboard models (Appendix D). On GSM8K, EssenceBench maintains strong agreement with full-leaderboard scores, with Pearson correlation $\geq 0.992$ across subset sizes (Table 13).

**Diversity, Bias, and Efficiency.** EssenceBench preserves diversity (change $\leq 2.3\%$; Table 14) and introduces negligible MMLU distribution shift (typically $< 2\%$; Table 15). It also achieves up to $49\times$ speedup on HellaSwag (Table 16). Details are reported in Appendix E.

## 4.4 QUALITATIVE ANALYSIS: REDUNDANCY

To demonstrate the effectiveness of our coarse filtering, we present representative cases of redundancy in Appendix F. We identify samples with high *textual similarity* (differing only by proper nouns or value placement) and *ranking similarity* (different narratives but identical reasoning structures). By filtering these effectively, EssenceBench maximizes the information density of the selected subset.

## 5 CONCLUSION

In this paper, we identified sample redundancy in LLM benchmark evaluation. To address this problem, we introduced EssenceBench, a coarse-to-fine benchmark compression framework that combines redundancy-aware filtering with an iterative genetic algorithm optimized for accurate reconstruction. Extensive experiments on five standard benchmarks show that EssenceBench achieves a $200\times$ reduction in benchmark size while preserving ranking fidelity.

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

# A  PSEUDO-CODE OF ESSENCEBENCH

We provide the detailed pseudo-code for the two core components of EssenceBench: the Genetic Algorithm (GA) for subset selection and the coarse-to-fine Iterative Refinement framework.

---

**Algorithm 1: Genetic Algorithm (Subset Selection)**

---

**Require:** Filtered Score Matrix $\mathbf{S} \in \{0, 1\}^{N_{\text{LLM}} \times M}$, Ground Truth $\mathbf{y} \in \mathbb{R}^{n_{\text{LLM}}}$, Target Size $k$, Population $N_{\mathcal{P}}$, Elites $N_{\mathcal{E}}$, Generations $N_G$.
**Ensure:** Best mask $\mathbf{m}^*$, Lowest error $\varepsilon^*$, Top elites $\mathcal{E}$
 1: Initialize population $\mathcal{P} = \{\mathbf{m}^{(i)}\}_{i=1}^{N_{\mathcal{P}}}$ with random binary $k$-masks
 2: $\mathbf{m}^* \leftarrow \mathbf{0}$, $\varepsilon^* \leftarrow +\infty$
 3: **for** $t = 1$ to $N_G$ **do**
 4:   **for** each $\mathbf{m} \in \mathcal{P}$ **do**
 5:     Estimate accuracy: $\hat{\mathbf{y}} = g(\mathbf{S} \cdot \mathbf{m}/k)$ {Using GAM predictor $g(\cdot)$}
 6:     Compute RMSE: $\varepsilon(\mathbf{m}) = \sqrt{\frac{1}{N_{\text{LLM}}} \sum (\hat{y}_i - y_i)^2}$
 7:     Fitness $F(\mathbf{m}) = -\varepsilon(\mathbf{m})$
 8:   **end for**
 9:   $\mathcal{E} \leftarrow$ Top $N_{\mathcal{E}}$ masks sorted by fitness
10:   $\mathcal{P}' \leftarrow \mathcal{E}$ {Elitism}
11:   **while** $|\mathcal{P}'| < N_{\mathcal{P}}$ **do**
12:     $\mathbf{m}_a, \mathbf{m}_b \leftarrow$ TournamentSelection($\mathcal{P}$)
13:     $\mathbf{m}_{child} \leftarrow$ Crossover($\mathbf{m}_a, \mathbf{m}_b$)
14:     $\mathbf{m}_{child} \leftarrow$ Mutate($\mathbf{m}_{child}$)
15:     $\mathbf{m}_{child} \leftarrow$ AdjustSize($\mathbf{m}_{child}, k$) {Ensure exactly $k$ ones}
16:     $\mathcal{P}' \leftarrow \mathcal{P}' \cup \{\mathbf{m}_{child}\}$
17:   **end while**
18:   $\mathcal{P} \leftarrow \mathcal{P}'$
19:   **if** $\min \varepsilon(\mathcal{P}) < \varepsilon^*$ **then**
20:     Update $\mathbf{m}^*$ and $\varepsilon^*$
21:   **end if**
22: **end for**
23: **return** $\mathbf{m}^*$, $\varepsilon^*$, $\mathcal{E}$

---

**Algorithm 2: Iterative Refinement with Attribution Guidance**

---

**Require:** Full Scores $\mathbf{S}$, Ground Truth $\mathbf{y}$, Target $k$, Max Rounds $R_{\max}$, Attr. Ratio $\alpha$.
**Ensure:** Optimal subset mask $\mathbf{m}^\star$
 1: Candidate Indices $\mathcal{I} \leftarrow \{1, \ldots, M\}$ (All samples)
 2: $\varepsilon^\star \leftarrow \infty$
 3: **for** $r = 0$ to $R_{\max} - 1$ **do**
 4:   {Step 1: Run GA on current pool}
 5:   $(\mathbf{m}, \varepsilon, \mathcal{E}) \leftarrow$ RUNGA($\mathbf{S}[:, \mathcal{I}], \mathbf{y}, k$)
 6:   **if** $\varepsilon < \varepsilon^\star$ **then**
 7:     $\mathbf{m}^\star \leftarrow \mathbf{m}$, $\varepsilon^\star \leftarrow \varepsilon$
 8:   **end if**
 9:   **if** $|\mathcal{I}| \leq k$ **or** $r = R_{\max} - 1$ **then**
10:     **break**
11:   **end if**
12:   {Step 2: Attribution-based Grouping}
13:   Compute Attribution $A_j$ for each sample $j \in \mathcal{I}$ using Elites $\mathcal{E}$
14:   Partition $\mathcal{I}$ into groups based on $A_j$: $G_{\text{High}}, G_{\text{Low}}, G_{\text{Rand}}$
15:   {Step 3: Select best group for next round}
16:   Evaluate candidate groups via fast GA search
17:   $\mathcal{I} \leftarrow \arg\min_G \varepsilon(G)$
18: **end for**
19: **return** $\mathbf{m}^\star$

---

## B FULL BENCHMARK RESULTS

We provide the complete evaluation tables for HellaSwag and ARC. These results complement the GSM8K results provided in the main text. EssenceBench demonstrates consistent superiority in both Score Reconstruction (RMSE) and Ranking Fidelity.

Table 6: Full Results on HellaSwag. Comparison of MetaBench vs. EssenceBench across all metrics. Highlighted rows indicate our method.

| Method | Metric | 50 | 100 | 150 | 200 | 250 | 300 | 350 | 400 | 450 | 500 |
|--------|--------|-----|-----|-----|-----|-----|-----|-----|-----|-----|-----|
| MetaBench | RMSE ↓ | 2.397 | 1.734 | 1.495 | 1.318 | 1.166 | 1.076 | 0.961 | 0.929 | 0.880 | 0.836 |
| | Pearson ↑ | 0.992 | 0.996 | 0.997 | 0.998 | 0.998 | 0.998 | 0.999 | 0.999 | 0.999 | 0.999 |
| | Kendall ↑ | 0.879 | 0.915 | 0.928 | 0.937 | 0.942 | 0.946 | 0.955 | 0.954 | 0.957 | 0.959 |
| | Top-50 Acc ↑ | 0.920 | **0.940** | **0.960** | 0.940 | 0.900 | 0.960 | 0.900 | 0.960 | 0.940 | 0.980 |
| | Rank Err (within 10%) ↑ | 0.902 | 0.961 | 0.983 | 0.989 | 0.991 | **1.000** | 0.998 | **1.000** | 0.998 | **1.000** |
| EssenceBench | RMSE ↓ | **2.153** | **1.453** | **1.038** | **0.863** | **0.706** | **0.635** | **0.504** | **0.461** | **0.409** | **0.419** |
| | Pearson ↑ | **0.994** | **0.997** | **0.999** | **0.999** | **0.999** | **0.999** | **1.000** | **1.000** | **1.000** | **1.000** |
| | Kendall ↑ | **0.899** | **0.931** | **0.951** | **0.959** | **0.966** | **0.967** | **0.973** | **0.974** | **0.978** | **0.978** |
| | Top-50 Acc ↑ | **0.960** | 0.920 | 0.940 | **0.940** | **0.940** | **0.980** | **0.960** | **0.960** | **0.960** | 0.980 |
| | Rank Err (within 10%) ↑ | **0.946** | **0.988** | **0.998** | **1.000** | **1.000** | **1.000** | **1.000** | **1.000** | **1.000** | **1.000** |

Table 7: Full Results on ARC. Comparison of MetaBench vs. EssenceBench across all metrics. Highlighted rows indicate our method.

| Method | Metric | 50 | 100 | 150 | 200 | 250 | 300 | 350 | 400 | 450 | 500 |
|--------|--------|-----|-----|-----|-----|-----|-----|-----|-----|-----|-----|
| MetaBench | RMSE ↓ | 2.968 | 2.082 | 1.690 | 1.511 | 1.332 | 1.186 | 1.077 | 0.961 | 0.890 | 0.836 |
| | Pearson ↑ | 0.986 | 0.993 | 0.995 | 0.996 | 0.997 | 0.998 | 0.998 | 0.998 | 0.999 | 0.999 |
| | Kendall ↑ | 0.870 | 0.898 | 0.922 | 0.928 | 0.936 | 0.942 | 0.949 | 0.955 | 0.959 | 0.961 |
| | Top-50 Acc ↑ | 0.900 | 0.880 | 0.900 | 0.900 | 0.880 | 0.880 | 0.920 | 0.920 | 0.900 | **0.960** |
| | Rank Err (within 10%) ↑ | 0.881 | 0.929 | 0.973 | 0.985 | 0.985 | 0.992 | 0.998 | 1.000 | 1.000 | 0.998 |
| EssenceBench | RMSE ↓ | **2.045** | **1.104** | **0.841** | **0.703** | **0.612** | **0.529** | **0.501** | **0.422** | **0.368** | **0.347** |
| | Pearson ↑ | **0.993** | **0.998** | **0.999** | **0.999** | **0.999** | **1.000** | **1.000** | **1.000** | **1.000** | **1.000** |
| | Kendall ↑ | **0.907** | **0.947** | **0.958** | **0.964** | **0.969** | **0.973** | **0.974** | **0.978** | **0.981** | **0.983** |
| | Top-50 Acc ↑ | **0.900** | **0.940** | **0.960** | **0.920** | **0.960** | **1.000** | **0.960** | **0.980** | **0.960** | 0.940 |
| | Rank Err (within 10%) ↑ | **0.949** | **0.997** | **1.000** | **1.000** | **1.000** | **1.000** | **1.000** | **1.000** | **1.000** | **1.000** |

## C DETAILED ANALYSIS OF SENSITIVITY

### C.1 HYPERPARAMETER SENSITIVITY (COARSE FILTERING)

We investigate the sensitivity of thresholds $\tau_{\text{text}}$ and $\tau_{\text{ranking}}$ used in the Coarse Filtering stage. Tables 8 and 9 show RMSE performance on GSM8K and ARC ($k = 50$). The variations are minimal ($\pm \sim 0.15$), indicating robustness to threshold selection.

Table 8: Sensitivity of $\tau_{\text{text}}$ and $\tau_{\text{ranking}}$ on GSM8K.

| $\tau_{\text{txt}} \backslash \tau_{\text{rank}}$ | 0.80 | 0.85 | 0.90 | 0.95 |
|------|------|------|------|------|
| 0.80 | 2.65 | 2.56 | 2.64 | 2.80 |
| 0.85 | 2.68 | 2.64 | 2.69 | 2.61 |
| 0.90 | 2.74 | 2.70 | 2.76 | 2.77 |
| 0.95 | 2.61 | 2.75 | 2.65 | 2.69 |

Table 9: Sensitivity of $\tau_{\text{text}}$ and $\tau_{\text{ranking}}$ on ARC.

| $\tau_{\text{txt}} \backslash \tau_{\text{rank}}$ | 0.80 | 0.85 | 0.90 | 0.95 |
|------|------|------|------|------|
| 0.80 | 2.04 | 2.14 | 2.00 | 2.06 |
| 0.85 | 2.10 | 2.03 | 2.11 | 2.03 |
| 0.90 | 2.09 | 2.18 | 2.05 | 2.16 |
| 0.95 | 2.05 | 2.13 | 2.06 | 2.06 |

## C.2 GENETIC ALGORITHM HYPERPARAMETERS

Tables 10 and 11 display the sensitivity to Population Size ($N_p$) and Elite Size ($N_\epsilon$). The method remains stable across a wide range of configurations.

Table 10: Sensitivity to GA Params on GSM8K.

| $N_p \backslash N_\epsilon$ | 10 | 20 | 30 | 40 |
|---|---|---|---|---|
| 100 | 2.74 | 2.78 | 2.72 | 2.64 |
| 200 | 2.73 | 2.67 | 2.63 | 2.71 |
| 300 | 2.69 | 2.64 | 2.66 | 2.66 |
| 400 | 2.60 | 2.66 | 2.70 | 2.62 |

Table 11: Sensitivity to GA Params on ARC.

| $N_p \backslash N_\epsilon$ | 10 | 20 | 30 | 40 |
|---|---|---|---|---|
| 100 | 2.18 | 2.22 | 2.09 | 2.14 |
| 200 | 2.08 | 2.05 | 2.13 | 2.06 |
| 300 | 2.05 | 2.21 | 2.07 | 2.09 |
| 400 | 2.14 | 2.13 | 2.08 | 2.04 |

## C.3 ROBUSTNESS TO DUPLICATE FILTERING STRATEGY

We compared our default sequential filtering against Random and Conservative ($\geq 2$ duplicates) strategies. As shown in Table 12, the choice of filtering strategy has negligible impact on final performance.

Table 12: Robustness of Duplication Filtering Strategies.

| Filtering Strategy | GSM8K | | ARC | |
|---|---|---|---|---|
| | $k = 50$ | $k = 100$ | $k = 50$ | $k = 100$ |
| Original (Sequential) | 2.758 | 1.565 | 2.017 | 1.265 |
| Random (Stochastic) | 2.760 | 1.469 | 2.101 | 1.312 |
| Conservative ($\geq 2$ duplicates) | 2.651 | 1.558 | 2.099 | 1.300 |

## D DETAILED ANALYSIS OF GENERALIZABILITY

In our original experimental setup, we have already implemented a rigorous data split protocol to evaluate generalizability. Specifically, for both the training and testing sets, the dataset is first ranked by score and then partitioned into ten equipotent strata. Within each stratum, 10% of the instances are stratifiedly sampled and subsequently pooled to constitute the test set; the remaining 90% are retained as the training set.

To further explore the generalizability on train-val-test partition, we conducted a rigorous hold-out experiment where we randomly selected **600 models among total 6736 models** from the leaderboard results as the test set, while using the remaining models for training and validation.

Our comprehensive results in Table 13 demonstrate that the compressed benchmarks exhibit **strong generalization** to unseen models. The consistently high correlation coefficients ($> 0.99$), ranking metrics, and error reduction patterns across all k-values prove the **generalizability** of EssenceBench.

Table 13: Generalizability on Unseen Models (GSM8K). Full results across metrics and subset sizes.

| Metric | 50 | 100 | 150 | 200 | 250 | 300 | 350 | 400 |
|---|---|---|---|---|---|---|---|---|
| RMSE $\downarrow$ | 3.463 | 2.323 | 1.937 | 1.703 | 1.499 | 1.323 | 1.327 | 1.122 |
| Pearson $\uparrow$ | 0.992 | 0.996 | 0.998 | 0.998 | 0.999 | 0.999 | 0.999 | 0.999 |
| Kendall $\uparrow$ | 0.881 | 0.914 | 0.932 | 0.942 | 0.950 | 0.953 | 0.957 | 0.962 |
| Top-50 Acc $\uparrow$ | 0.360 | 0.600 | 0.740 | 0.820 | 0.940 | 0.880 | 0.940 | 0.920 |
| Rank Err (within 10%) $\uparrow$ | 0.923 | 0.957 | 0.982 | 0.990 | 0.995 | 0.993 | 0.995 | 0.998 |

## E  DETAILED ANALYSIS OF DIVERSITY, BIAS AND EFFICIENCY

**Diversity.**  We evaluate whether compression reduces data diversity on GSM8K, ARC, and HellaSwag. Following prior work, we embed each text sample $x_i$ into a semantic vector $\mathbf{v}_i$ using BGE-M3, and measure pairwise cosine similarity $\mathrm{sim}(\mathbf{v}_i, \mathbf{v}_j)$. We define the diversity score as the complement of the average pairwise similarity:

$$\mathcal{D} = 1 - \frac{1}{N(N-1)} \sum_{i=1}^{N} \sum_{j \neq i} \mathrm{sim}(\mathbf{v}_i, \mathbf{v}_j), \tag{13}$$

where $N$ is the number of samples. Higher $\mathcal{D}$ indicates greater diversity. As shown in Table 14, EssenceBench preserves diversity well, with relative changes within 2.3% across all three benchmarks.

Table 14: Diversity scores $\mathcal{D}$ (complement of average cosine similarity). Higher is more diverse.

| Method | GSM8K | ARC | HellaSwag |
|---|---|---|---|
| Full Dataset | 0.397 | 0.367 | 0.438 |
| EssenceBench | 0.393 | 0.366 | 0.428 |
| Relative Change | -1.01% | -0.27% | -2.28% |

**Bias (Distributional Shift).**  We further analyze potential selection bias on MMLU by comparing the subject distribution between the full dataset $\mathcal{D}_{\mathrm{raw}}$ and the selected subset $\mathcal{D}_{\mathrm{sel}}$ with fixed subset size $N = 2000$. Let $P_{\mathrm{raw}}(i)$ and $P_{\mathrm{sel}}(i)$ denote the percentage of subject $i$ in $\mathcal{D}_{\mathrm{raw}}$ and $\mathcal{D}_{\mathrm{sel}}$, respectively. We quantify distributional shift using the deviation ratio:

$$\Delta_{\mathrm{ratio}}(i) = \frac{|P_{\mathrm{sel}}(i) - P_{\mathrm{raw}}(i)|}{P_{\mathrm{raw}}(i)}. \tag{14}$$

Lower $\Delta_{\mathrm{ratio}}$ indicates better preservation of the original distribution. Table 15 reports six representative subjects and shows consistently small deviations (typically $< 2\%$), suggesting that EssenceBench introduces negligible selection bias.

Table 15: Subject distributional shift on MMLU ($N = 2000$). $\Delta_{\mathrm{ratio}}$ measures relative deviation.

| Subject | Raw (%) | EssenceBench (%) | $\Delta_{\mathrm{ratio}}$ |
|---|---|---|---|
| Elementary Mathematics | 2.722 | 2.725 | 0.0010 |
| College Medicine | 1.260 | 1.247 | 0.0101 |
| International Law | 0.857 | 0.872 | 0.0176 |
| Management | 0.756 | 0.743 | 0.0182 |
| Econometrics | 0.807 | 0.822 | 0.0187 |
| College Chemistry | 0.706 | 0.721 | 0.0212 |

**Efficiency.**  We quantify the efficiency by adding all the time together. Let $T_{\mathrm{full}}$ denote the time of full-benchmark evaluation, $T_{\mathrm{comp}}$ the compression cost, and $T_{\mathrm{eval}}$ the evaluation time on the compressed subset. The evaluation then costs $T_{\mathrm{comp}} + T_{\mathrm{eval}}$. Table 16 reports the speedups.

Table 16: Computational Cost Analysis and Acceleration Factors. Time is measured in hours.

| Benchmark | Size | Full Eval. Time (h) | EssenceBench Cost (Hours) Comp. | Eval. | Total | Acceleration ($\uparrow$) |
|---|---|---|---|---|---|---|
| GSM8K | 1319 | 1.96 | 0.27 | 0.07 | 0.34 | 4.76× |
| ARC | 1172 | 1.97 | 0.29 | 0.08 | 0.37 | 4.32× |
| HellaSwag | 10,003 | 22.06 | 0.33 | 0.11 | 0.44 | 49.14× |
| WinoGrande | 1267 | 0.78 | 0.28 | 0.03 | 0.31 | 1.52× |

# F    CASE STUDY

As clearly illustrated in Table 17, the proposed filtering mechanism successfully and consistently identifies semantically equivalent items across various cases. In the text redundancy case, two questions differ in phrasing but share the same arithmetic structure. In the ranking redundancy case, two problems with different narratives yield similar model scores because both problems require multi-step numerical reasoning that includes proportional calculations, intermediate variable derivation, and aggregation of weighted quantities. These representative cases provide clear evidence that EssenceBench effectively captures and distinguishes both superficial and deeper structural redundancies.

Table 17: Examples of Redundancy handled by EssenceBench. Orange and Teal highlight identical structures or reasoning patterns.

| Textual Redundancy (Lexical Overlap) | Ranking Redundancy (Score Correlation) |
|---|---|
| Zack's locker is half as big as Timothy's locker. Peter's locker is 1/4 as big as Zack's locker. If Peter's locker is 5 cubic inches, how big is Timothy's locker in cubic inches? | Axel has 50 silver pesos and 80 gold pesos. He visits her friend Anna who has twice as many silver pesos as he has and 40 more gold pesos. What's **the total number** of pesos they have together? |
| Timothy's locker is 24 cubic inches. Zack's locker is half as big as Timothy's locker. Peter's locker is 1/4 as big as Zack's locker. How big is Peter's locker in cubic inches? | Amy is taking a history test. She correctly answers 80% of the multiple-choice questions, 90% of the true/false questions, and 60% of the long-answer questions. The multiple-choice and true/false questions are worth 1 point each, and the long answer questions are worth 5 points each. How many **points does Amy score** if there are 10 multiple-choice questions, 20 true/false questions, and 5 long answer questions? |

