# OpenReview forum: "Rethinking LLM Evaluation: Can We Evaluate LLMs with 200× Less Data?"
_ICLR.cc/2026/Conference — ICLR 2026 Poster_

### Official Review · Reviewer_TShU · 2025-10-29

**Soundness:** 2
**Presentation:** 2
**Contribution:** 2
**Rating:** 4
**Confidence:** 4

**Summary:**

The paper proposes EssenceBench, a coarse‑to‑fine framework for compressing LLM benchmarks so that rankings and overall scores can be reconstructed with far fewer evaluation items. The pipeline (i) performs coarse filtering to remove redundant items using text similarity (embedding cosine) and ranking similarity (correlation of model outcomes), (ii) runs a Genetic Algorithm to select subsets that minimize a GAM‑predicted reconstruction error (RMSE) of full‑set accuracies across models, and (iii) applies attribution‑guided selection using an Explainable Boosting Machine (EBM) to score per‑item contribution, group items (high/low/random attribution), and re‑search within the best group; this is iterated to improve diversity and convergence. Experiments on five benchmarks (GSM8K, ARC, HellaSwag, WinoGrande, MMLU) report lower RMSE and better rank preservation than MetaBench and other baselines, with large reported reductions (e.g., 25×–200× fewer samples on HellaSwag) and faster compression.

**Strengths:**

1. Casting benchmark compression as minimizing an error between full‑set and subset‑based scores is crisp and aligns the objective with leaderboard reconstruction
2. The paper quantifies both text redundancy and ranking redundancy and visually demonstrates non‑trivial redundancy on popular datasets
3. The three‑stage design is easy to follow and practically implementable.

**Weaknesses:**

1. The duplication filter “keeps the earlier item and discards the later one if similarity exceeds thresholds”, which introduces order bias and potential instability across random permutations of the dataset. No robustness check to ordering is reported.
2. The paper mentions bge‑m3 embeddings and ranking correlations, but concrete threshold values, normalization, and sensitivity are absent
3. Fig 2C shows wall clock hours, but the hardware, eval setup is not clear. How do you compare the eval time?

**Questions:**

1. In Section 3.3, the fitness uses a GAM g that maps subset accuracy s_j to full‑set accuracy y_j. It is unclear whether g is retrained per candidate subset or pre‑trained on a pool of subsets. The compute/variance implications differ drastically and affect the efficiency claim.

---

> ### Author Response · Authors · 2025-11-21
> **(Part 1) Response to Reviewer TShU**
>
> ### Q1: About the ordering robustness in duplication filtering.
>
> We appreciate the reviewer raising this important question regarding potential order sensitivity in our duplication filtering process.
>
> **Method Design.** We attribute the stability of our approach to the fundamental design of the Coarse Filtering stage. Its primary function is to prune highly redundant information to accelerate the search process, rather than to make final selection decisions. Conceptually, whether the filter retains "Sample A" or its semantic equivalent "Sample B," the **informational content** of the candidate pool remains effectively unchanged. The subsequent Genetic Algorithm is highly adaptive and capable of extracting the necessary signal from the remaining pool, regardless of which specific duplicate was preserved. Consequently, the efficient index-based filtering used in our paper is both sufficient and robust.
>
> **Additional Experimental Results.** To verify this hypothesis empirically, we conducted an ablation study comparing our default strategy against alternative mechanisms that simulate data permutation. We evaluated these strategies on both the GSM8K and ARC benchmarks across various target subset sizes (denoted as $k$).
>
> Specifically, we compared three filtering implementations:
> 1.  **Original:** When a pair of samples exhibits high ranking similarity, we deterministically retain the sample with the smaller index (i.e., the earlier item).
> 2.  **Random:** When a highly similar pair is identified, we randomly select one to retain and discard the other. This simulates the effect of shuffling the dataset order.
> 3.  **Conservative:** A stricter criterion where a sample is discarded only if it shows high similarity with at least two other samples, rather than just one.
>
>
> **GSM8K Benchmark (RMSE $\downarrow$)**
> | Filtering Strategy | $k=50$ | $k=100$ |
> | :--- | :---: | :---: |
> | Original (Sequential) | 2.7576 | 1.5646 |
> | Random (Stochastic) | 2.7604 | 1.4687 |
> | Conservative ($\ge 2$ duplicates) | 2.6509 | 1.5576 |
>
> **ARC Benchmark (RMSE $\downarrow$)**
> | Filtering Strategy | $k=50$ | $k=100$ |
> | :--- | :---: | :---: |
> | Original (Sequential) | 2.0171 | 1.2648 |
> | Random (Stochastic) | 2.1014 | 1.3120 |
> | Conservative ($\ge 2$ duplicates) | 2.0988 | 1.2998 |
>
> As shown in the results above, the performance differences are minimal, showing that filtering strategy having negligible impact on final performance. **We have followed your advice to add a new section in Appendix H (Table 11-12) for this discussion.**
>
> ### Q2: About hyper-parameters in bge-m3 embedding.
>
>
> Thanks for your question. All parameters are set in accordance with https://huggingface.co/BAAI/bge-m3/blob/main/README.md, where batch_size=12, max_length=8192, and fp16 precision for acceleration, with no other special settings applied. We have followed your advice to add detailed settings in the revised manuscript.

---

> ### Author Response · Authors · 2025-11-21
> **(Part 2) Response to Reviewer TShU**
>
> ### Q3: About details of Figure2.
>
> To clarify, Figure 2C depicts the wall-clock time required by both methods to compress the dataset, rather than directly evaluating all the models for all benchmarks. Specifically, we extracted the archieved results from Open LLM Leaderboard (https://huggingface.co/datasets/open-llm-leaderboard-old/results), which has the result of each instance of each benchmark for thousands of models. These compression  were conducted on identical hardware equipped with 8x NVIDIA H200 GPUs and are distinct from model evaluation tasks.  We have updated the text in the Introduction to more accurately describe Figure 2.
>
> ### Q4: About implementation details of GAM.
>
>
> We appreciate the reviewer's careful examination of Section 3.3 regarding the training schedule of the Generalized Additive Model (GAM). We clarify that the GAM is **not retrained for each candidate subset**. Instead, it is trained **once at the beginning of the subset selection phase** and serves as a **fixed, static surrogate** throughout the entire Genetic Algorithm (GA) search process.
>
> We implement this via **per-round surrogate fitting**. At the start of each GA round, we fit a single GAM, denoted as $g_R$, on the current feature pool. This model remains fixed while the GA explores the search space. For every candidate mask $\mathbf{m}$ generated during the evolutionary process, its fitness is evaluated via a single forward pass through this surrogate:
> $$ \mathrm{fit}(\mathbf{m}) = -\mathrm{RMSE}\bigl(\mathbf{y},g_R(\mathbf{S}_{\mathbf{m}})\bigr) $$
> Because $g_R$ is trained only once per round and reused, the computational cost is effectively limited to "one training step plus many cheap inferences."
>
> This design is critical for both efficiency and stability. By avoiding per-candidate retraining, we ensure that the heavy computational overhead of model fitting is not incurred during the search loop. Furthermore, evaluating all candidates against the same fixed surrogate ensures that all masks in a given round lie on the same fitness surface, eliminating the potential for statistical variance that would arise from retraining the predictor for each individual subset.
>
> **We have followed your advice to present this training schedule and its efficiency implications more clearly in Section 3.3 of the revised paper.**

---

> ### Author Response · Authors · 2025-11-26
>
> Dear Reviewer TShU,
>
>
> Thank you once again for your valuable comments on our submission. As the discussion phase is approaching its end, we would like to kindly confirm whether we have sufficiently addressed all of your concerns (or at least part of them). Should there be any remaining questions or areas requiring further clarification, please do not hesitate to let us know. If you are satisfied with our responses, we would greatly appreciate your consideration in adjusting the evaluation scores accordingly.
>
>
> We sincerely look forward to your feedback.

---

### Official Review · Reviewer_MmMV · 2025-10-31

**Soundness:** 3
**Presentation:** 3
**Contribution:** 3
**Rating:** 4
**Confidence:** 3

**Summary:**

For addressing the growing cost and redundancy of large-scale LLM benchmarking, the study presents a benchmark compression framework, EssenceBench. It aims to preserve ranking fidelity while substantially reducing evaluation cost for large language models. The approach combines redundancy-aware filtering with a genetic algorithm and attribution-guided refinement to identify representative subsets of benchmark data. Experimental results on five datasets show consistent improvements over prior methods. It highlights the potential of EssenceBench to make large-scale LLM evaluation more efficient and scalable.

**Strengths:**

1. The proposed method addresses an increasingly important and practical challenge in LLM evaluation concerning benchmark cost without losing ranking reliability.
2. The integration of fitness-based subset selection with attribution-based sample selection provides an effective paradigm for benchmark compression while maintaining the intrinsic properties of datasets (e.g., diversity).
3. Experimental results across five benchmarks and multiple baselines demonstrate the strong empirical performance of EssenceBench.

**Weaknesses:**

1. The sample redundancy phenomenon has been widely observed in prior studies on dataset quality assessment. The paper should more clearly differentiate EssenceBench from these existing efforts to highlight the necessity of Genetic Algorithms.
2. The study lacks the computational or time complexity analysis of the proposed framework, which may better demonstrate the feasibility of EssenceBench for datasets with different scales.
3. The parameter sensitivity analysis is limited, which only investigates the effects of the number of generations (gens) and refinement rounds (rounds). Other key hyperparameters (e.g. $\tau_{text}$, $\tau_{ranking}$ and $N_{\mathcal{P}}$) that likely have a substantial impact on compression quality are omitted. It raises the concern regarding parameter search complexity and the overall robustness of the proposed framework.
4. The experimental evaluation narrowly focuses on the extrinsic metric of prediction error (RMSE) and ranking consistency. The experiments lack some quantitative metric to evaluate intrinsic quality and characteristics of the compressed datasets (e.g., diversity, bias and so on).

**Questions:**

1. Could the evaluation be extended with objective metrics that assess the intrinsic properties of the compressed datasets (e.g., diversity [1], bias [2] and so on)
2. An analysis of the genetic algorithm's computational complexity and a justification for the number of generations required for convergence are needed.
3. A more comprehensive sensitivity analysis is required for the hyperparameters to verify the robustness of the proposed framework, along with a discussion on the parameter search strategy.
4. The definition 4 lacks sufficient details about how the ranking scores $\mathbf{r}$ are computed from several LLMs.
5. The case study in Table 3 offers an interesting glimpse, but a broader analysis is needed to fully support the claims about model behavior. Could you include a dedicated appendix section with more examples across different datasets to better illustrate the duplication-aware filtering in EssenceBench?
6. In Figure 2, the legends of MetaBench and EssenceBench are mislabeled.
7. Part of notations are inconsistent. For example, Equation 12 defines $j \in \mathcal{I}(m)$ while the main text refers to $j \in \mathcal{I}(\mathcal{D}_{filtered})$

[1] On LLMs-Driven Synthetic Data Generation, Curation, and Evaluation: A Survey. 2024.

[2] Bias and fairness in large language models: A survey. 2024

---

> ### Author Response · Authors · 2025-11-21
> **(Part 1) Response to Reviewer MmMV**
>
> ### Q1: About the role of Genetic Algorithm.
>
> We thank the reviewer for this insightful comment. We agree that sample redundancy is a shared concern with prior dataset quality assessment research. However, our work differs fundamentally in its objective, i.e., Evaluation vs. Training, and its methodological necessity, i.e., Global Optimization via GA vs. Local Filtering.
>
> **1. Difference between our Essencebench and previous works**. While prior studies focus on redundancy to improve *training efficiency*, i.e., coreset selection to maintain model performance with fewer data for a model, EssenceBench addresses redundancy to improve *evaluation efficiency* and *sensitivity*. Specifically, existing work Typically evaluates the "overall dataset quality" using proxy metrics like PPL or gradient norms to filter "easy" or "noisy" samples for training. In contrast, we define redundancy specifically in the context of model discrimination. A sample is "redundant" in a benchmark not just if it is textually similar, but if it fails to distinguish between models, e.g., all models answer it correctly, or the pass/fail pattern is identical to another question. We aim to select a subset that maximizes the divergence in model behaviors, which existing heuristic methods do not optimize for.
>
> **2. Necessity of Genetic Algorithms (GA)**. Standard filtering methods (e.g., removing samples with low PPL) operate on. a *sample-wise* basis and cannot capture the collective discriminative power of a subset. For most coreset selection methods, the objective is to prune redundant samples from a single model's perspective, rather than all models.
>
> Mathematically, we employ GA because the selection of an optimal benchmark subset is a combinatorial **NP-hard problem**. Specifically, our optimization goal (maximizing the separation or ranking correlation between models) is non-differentiable, making gradient-based methods inapplicable. Besides, greedy approaches or simple heuristics (like PPL/uncertainty thresholds) often select locally optimal samples but fail to find the combination of samples that best covers the diverse capabilities of LLMs. GA allows us to search the global solution space to find the "essence" subset that provides the most information gain.
>
> ### Q2: About computational efficiency.
>
> Thanks. Accordingly, we conducted the computational cost experiments in GSM8K, ARC, Hellaswag, WinoGrande, and the results are as follows:
> | Benchmark | Full Size | Full Eval. Time (Hours) | Compression Cost (Hours) | Compressed Eval. Time (Hours) | Total Time for EssenceBench (Hours) | Acceleration |
> |---|---|---|---|---|---|---|
> | GSM8k | 1,319 | 1.96 | 0.27 | 0.07 | 0.34 | 4.76x |
> | ARC | 1,172 | 1.97 | 0.29 | 0.08 | 0.37 | 4.32x |
> | HellaSwag | 10,003 | 22.06 | 0.33 | 0.11 | 0.44 | 49.14x |
> | WinoGrande | 1,267 | 0.78 | 0.28 | 0.03 | 0.31 | 1.52x |
>
> All experiments were conducted on a single H200 GPU using the Qwen3-8B model. Inference was performed using the standard transformers library with a batch size of 1.  The results from this table is that **the break-even point is less than one for all benchmarks**. This means that for a practitioner evaluating even a single model, the total time spent (i.e., the one-time compression cost and the first compressed evaluation) is already significantly less than the time it would have taken to run that single evaluation on the full benchmark. Notably, our method has minimal GPU requirements, needing only about 2GB of VRAM per task. This demand is low enough to be met even by personal laptops. **We have added discussion for this efficiency in Table 15, Appendix K.**

---

> ### Author Response · Authors · 2025-11-21
> **(Part 2) Response to Reviewer MmMV**
>
> ### Q3: About additional hyper-parameter sensitivity.
>
> #### Q3.1 About GA algorithm hyper-parameters sensitivity.
> Thank you for this valuable feedback. In response, we have conducted extensive experiments to evaluate the sensitivity of key genetic algorithm parameters, including population size $N_p$ and elite size $N_ε$, across different compression scales.
>
> **Ablation results on population size $N_p$ and elite size $N_ε$:**
>
> GSM8K benchmark results (target size k=50, RMSE $\downarrow$):
> | $N_p$ \ $N_ε$ | 10 | 20 | 30 | 40 |
> |------------|-----|-----|-----|-----|
> | **100** | 2.7375 | 2.7793 | 2.7240 | 2.6428 |
> | **200** | 2.7279 | 2.6745 | 2.6314 | 2.7058 |
> | **300** | 2.6917 | 2.6378 | 2.6611 | 2.6568 |
> | **400** | 2.5998 | 2.6639 | 2.7046 | 2.6227 |
>
> ARC benchmark results (target size k=50, RMSE $\downarrow$):
> | $N_p$ \ $N_ε$ | 10 | 20 | 30 | 40 |
> |------------|-----|-----|-----|-----|
> | **100** | 2.1790 | 2.2165 | 2.0861 | 2.1425 |
> | **200** | 2.0803 | 2.0509 | 2.1256 | 2.0584 |
> | **300** | 2.0494 | 2.2141 | 2.0741 | 2.0856 |
> | **400** | 2.1413 | 2.1271 | 2.0788 | 2.0448 |
>
> **Summary of Results.** As shown in the tables above, we find that performance variations across different population sizes (100-400) are minimal. Besides, elite sizes from 10 to 40 show consistent performance across both benchmarks.
>
> Therefore, our comprehensive analysis demonstrates that the proposed framework exhibits remarkable robustness to key genetic algorithm hyperparameters. The minimal performance variations across different population sizes and elite sizes directly address the reviewer's concern about parameter search complexity. **We have followed your advice to add a new section in Appendix G (Table 9 and Table 10) for this discussion.**
>
> #### Q3.2: Ablation Study on $\tau_{\mathrm{text}}$ and $\tau_{\mathrm{ranking}}$.
>
> To systematically investigate this, we conducted comprehensive sensitivity analyses across two benchmarks: GSM8K (mathematical reasoning) and ARC (scientific QA). We evaluated the robustness with target subset size $k=50$.
>
>
>
> **Results on GSM8K Benchmark (k=50, RMSE $\downarrow$)**
> | $\tau_{\mathrm{text}}$ \ $\tau_{\mathrm{ranking}}$ | 0.80 | 0.85 | 0.90 | 0.95 |
> |---------------------|------|------|------|------|
> | **0.80** | 2.6548 | 2.5577 | 2.6380 | 2.8047 |
> | **0.85** | 2.6781 | 2.6352 | 2.6857 | 2.6079 |
> | **0.90** | 2.7433 | 2.6971 | 2.7562 | 2.7697 |
> | **0.95** | 2.6108 | 2.7499 | 2.6455 | 2.6877 |
>
> **Results on ARC Benchmark (k=50, RMSE $\downarrow$)**
> | $\tau_{\mathrm{text}}$ \ $\tau_{\mathrm{ranking}}$ | 0.80 | 0.85 | 0.90 | 0.95 |
> |---------------------|------|------|------|------|
> | **0.80** | 2.0353 | 2.1378 | 2.0043 | 2.0636 |
> | **0.85** | 2.1021 | 2.0307 | 2.1077 | 2.0270 |
> | **0.90** | 2.0939 | 2.1814 | 2.0496 | 2.1607 |
> | **0.95** | 2.0520 | 2.1293 | 2.0585 | 2.0608 |
>
> As shown in the table above, we find that across both benchmarks, performance variations remain within a narrow range ($\pm \sim 0.1-0.15$ points) across all threshold combinations, indicating low sensitivity. Therefore, our analysis demonstrates that the proposed framework exhibits **remarkable robustness** to key thresholds in the coarse filtering process. The minimal performance variations across different $\tau_{\mathrm{text}}$ and $\tau_{\mathrm{ranking}}$ directly address the reviewer's concern. **We have followed your advice to add a new section in Appendix F (Table 7 and Table 8) for this discussion.**

---

> ### Author Response · Authors · 2025-11-21
> **(Part 3) Response to Reviewer MmMV**
>
> ### Q4: About intrinsic properties of compressed datasets.
> Thank you for the references. Given the critical importance of ensuring bias consistency and diversity preservation across the compression pipeline, we compared the pre- and post-compression data metrics. The details are outlined as follows:
>
>
> #### 1. Diversity
>
> | Method | GSM8K | ARC | HellaSwag |
> | :--- | :--- | :--- | :--- |
> | **Raw** | 0.397 | 0.367 | 0.438 |
> | **EssenceBench** | 0.393 | 0.366 | 0.428 |
> | **Relative Change** | -1.01% | -0.27% | -2.28% |
>
> We evaluated data diversity across the **GSM8K, ARC, and Hellaswag** benchmarks. To quantify this metric, we utilized the **BGE-M3** embedding model to map each text sample $x_i$ into a semantic vector representation $\mathbf{v}_i$. We then computed the pairwise cosine similarity between these embeddings. The final diversity score $\mathcal{D}$ is defined as the complement of the average similarity, formulated as:
>
> $$ \mathcal{D} = 1 - \frac{1}{N(N-1)} \sum_{i=1}^{N} \sum_{j \neq i} \text{sim}(\mathbf{v}_i, \mathbf{v}_j) $$
>
> where $N$ denotes the number of samples, and $\text{sim}(\mathbf{v}_i, \mathbf{v}_j)$ represents the cosine similarity between vectors $\mathbf{v}_i$ and $\mathbf{v}_j$.
>
> It can be observed from the table that the Essencebench compressed data exhibits **negligible difference** in diversity relative to the original baseline.
>
> #### 2. Bias
>
> | Dataset                | Raw(%) | EssenceBench(%) | $\Delta_{ratio}$ |
> | :--------------------- | :----- | :-------------- | :--------------- |
> | college_chemistry      | 0.7056 | 0.7209          | 0.02117           |
> | econometrics           | 0.8065 | 0.8219          | 0.1865           |
> | international_law      | 0.8569 | 0.8723          | 0.1764           |
> | elementary_mathematics | 2.7218 | 2.7251          | 0.0010           |
> | college_medicine       | 1.2601 | 1.2472          | 0.1008           |
> | management             | 0.7560 | 0.7426          | 0.01815           |
>
> To investigate potential selection bias in EssenceBench, we conducted experiments on the MMLU benchmark. We fixed the target selection size at **$N = 2,000$** to analyze the distributional alignment between the selected subset and the original dataset. The table above presents the statistics for six randomly sampled subjects.
>
> We denote the full MMLU dataset as $\mathcal{D}\_{raw}$ and the subset selected by EssenceBench as $\mathcal{D}\_{sel}$, where $|\mathcal{D}\_{sel}| = N$. For each subject category $i$, we calculated the probability density (percentage) as the item count of that subject divided by the total dataset size. This yields two distributions:
>
> 1.  **Raw Distribution ($P_{raw}$):** The percentage of subject $i$ in the original MMLU dataset.
> 2.  **EssenceBench Distribution ($P_{sel}$):** The percentage of subject $i$ in the selected subset.
>
> To quantify the distributional shift, we define the distributional deviation ratio, denoted as $\Delta_{ratio}(i)$. This metric measures the relative divergence of the selected distribution from the original distribution, formulated as:
>
> $$
> \Delta_{ratio}(i) = \frac{|P_{sel}(i) - P_{raw}(i)|}{P_{raw}(i)}
> $$
>
> This ratio reflects the percentage change in the subject's representation relative to its original weight. A lower $\Delta_{ratio}$ indicates a higher fidelity to the original data distribution.
>
> As shown in the table, the values of $\Delta_{ratio}$ are consistently low across all sampled subjects (e.g., $0.10\\%$ for Elementary Mathematics). This indicates that EssenceBench maintains the proportional representation of each subject with high precision. Such minimal deviation suggests that our method **preserves the intrinsic diversity of the dataset while introducing negligible selection bias**.
>
>
> **We have also followed your advice to put the additional results in the Table 26 and Table 27 in Appendix N revised manuscript.**

---

> ### Author Response · Authors · 2025-11-21
> **(Part 4) Response to Reviewer MmMV**
>
> ### Q5: About details of Definition 4.
> We rewrite the definition 4 for better understanding ranking similarity:
>
> Let $\mathcal{D} = \left \\{ x\_1, x_2, \ldots, x_N \right \\} $ denote the benchmark dataset containing $N$ samples. To evaluate the ranking redundancy, we utilize a set of $K$ distinct Large Language Models (LLMs), denoted as $\mathcal{M} = \left \\{ m_1, m_2, \ldots, m_K \right\\}$.
>
> For each sample $x_i \in \mathcal{D}$, we construct a **performance vector** $\mathbf{s}_i \in \mathbb{R}^K$, where the $k$-th element represents the score (e.g., likelihood or accuracy) of model $m_k$ on sample $x_i$. The redundancy between two samples $x_i$ and $x_j$ is quantified by the Pearson correlation coefficient of their performance vectors:
> $$
>     \rho(\mathbf{s}_i, \mathbf{s}_j) = \frac{\text{Cov}(\mathbf{s}\_i, \mathbf{s}\_j)}{\sigma\_{\mathbf{s}\_i} \sigma\_{\mathbf{s}\_j}},
> $$
> where $\rho \in [-1, 1]$. A high correlation indicates that the set of models exhibits an identical performance pattern on both samples, implying they provide redundant information regarding model ranking capabilities.
>
> To prune the dataset, we define a redundancy threshold $\tau$ (set to 0.9 in our experiments). The set of redundant samples to be removed, $\mathcal{S}_{\text{remove}}$, is identified as:
>
> $$
>   \mathcal{S}\_{remove} = \left \\{  x_i \mid \exists j > i,  \text{s.t.}, \rho( \mathbf{s}_i, \mathbf{s}\_j) > \tau \right \\}
> $$
>
>
> **We have followed your advice to revised our presentation in Section 3.2 in the revised manuscript.**

---

> ### Author Response · Authors · 2025-11-21
> **(Part 5) Response to Reviewer MmMV**
>
> ### Q6: Ask for more case studies.
> We agree that a case study analysis provides compelling insights. Consequently, **we have included representative case studies for the Hellaswag, ARC, and Winogrande datasets in Figure 6-11 in Appendix O in this revision.**
>
> #### 1. Hellaswag
>
> | Text                                                         | Ranking                                                      |
> | ------------------------------------------------------------ | ------------------------------------------------------------ |
> | [header] How to organize pokémon cards [title] Organize by deck boxes. [step] This system works well if you are still building your collection and most of your cards are collected into decks. Simply place your cards into an empty deck box and place the box somewhere convenient, like on a shelf or in your closet. **A**. [substeps] Shipping your cards is also best, as others will drop the cards as they turn out so that the ones you put in are in their corresponding boxes. Do not make your cards exactly duplicates, however. **B**. [substeps] Keep the cards in a box in your bedroom, on your desk, or wherever else it is comfortable. You should also be able to easily remember which cards make up an entire deck. **C**. [substeps] This system trains your pokémon to flip to one deck box and then move on to another, switching between boxes. It may also function better to manually store the cards. **D**. [substeps] Since you've likely started to change your decks to be stronger when playing, you may want to name or label your decks so you know exactly what each box contains. For example, you might put a piece of tape on a box and write on the tape, psychic pokémon deck. | A girl in a sum suit in thrown from a wheel barrow and the referee counts the match as finished. we **A**. see the closing screen on black. **B**. see the ending title screen. **C**. then see an starting title screen. **D**. then see the olympian and a replay of the brawl: two players are handed out and one players left. |
> | [header] How to organize pokémon cards [title] Organize by deck boxes. [step] This system works well if you are still building your collection and most of your cards are collected into decks. Simply place your cards into an empty deck box and place the box somewhere convenient, like on a shelf or in your closet. **A**. This should be the most pleasing place for you to organize your cards, and to make your cards relatively easy to collect. [substeps] Remember to keep your deck box somewhere out of the way that shouldn't spread dust and mud, such as your bed, dresser, or counter top. **B**. Left-handed and right-handed pokémon cards are a good option for organizing with loose deck boxes. [substeps] Move your cards into boxes with open tabs that you can store your cards in during your games. **C**. [substeps] Go through your cuticles regularly to keep small, everyday piles from inadvertently consolidating into big piles. Set aside five large cards (two blue, seven pink, eight red , 13 blue , 7 large red , 7 medium red , 8 large pink, and 8 larger red) in each box. **D**. [substeps] Since you've likely started to change your decks to be stronger when playing, you may want to name or label your decks so you know exactly what each box contains. For example, you might put a piece of tape on a box and write on the tape, "" psychic pokémon deck. | A girl in a sum suit in thrown from a wheel barrow and the referee counts the match as finished. we **A**. see the closing screen on black. **B**. see the ending title screen. **C**. then see an starting title screen. **D**. then see the olympian and a replay of the brawl: two players are handed out and one players left. |

---

> ### Author Response · Authors · 2025-11-21
> **(Part 6) Response to Reviewer MmMV**
>
> #### 2. ARC
>
> | Text                                                         | Ranking                                                      |
> | ------------------------------------------------------------ | ------------------------------------------------------------ |
> | The human population is increasing every day. How does this increase in population affect the natural resources that maintain humans? **A**. increases the supply of resources **B**. decreases the amount of resources **C**. results in less demand for resources  **D**. improves the quality of the resources | A physicist wants to determine the speed a car must reach to jump over a ramp. The physicist conducts three trials. In trials two and three, the speed of the car is increased by 20 miles per hour. What is the physicist investigating when he changes the speed?  **A**. the control **B**. the hypothesis statement **C**. the dependent (responding) variable **D**. the independent (manipulated) variable' |
> | The human population is increasing every day. How does this increase in population affect the natural resources that maintain humans? **A**. It increases the supply of resources. **B**. It decreases the amount of resources. **C**. It results in less demand for resources. **D**. It improves the quality of the resources. | The end result in the process of photosynthesis is the production of sugar and oxygen. Which step signals the beginning of photosynthesis? **A**. Chemical energy is absorbed through the roots. **B**. Light energy is converted to chemical energy. **C**. Chlorophyll in the leaf captures light energy. **D**. Sunlight is converted into chlorophyll. |
>
> #### 3. Winogrande
>
> | Text                                                         | Ranking                                                      |
> | ------------------------------------------------------------ | ------------------------------------------------------------ |
> | The toddler kicked the soft furniture and hard toy, but only the _ caused pain. **A**. furniture **B**. toy | Leslie was nervous around parrots but Neil was not, since _ was bitten by a bird early in life. **A**. Leslie **B**. Neil |
> | The toddler kicked the hard furniture and soft toy, but only the _ caused pain. **A**. furniture **B**. toy | Christmas was a special holiday to Eric but not Adam since _ was a Jew. **A**. Eric **B**. Adam |
>
> These case studies reveal distinct characteristics regarding data similarity. In terms of textual similarity, the datasets exhibit extreme lexical overlap, often differing by only a few tokens. For instance, in the ARC dataset, options are distinguished solely by the addition of a subject while the remaining text is identical, highlighting a high degree of redundancy.
>
> Conversely, regarding ranking-oriented similarity, the data demonstrates higher-dimensional semantic complexity. As observed in the Winogrande dataset, relying on syntactic rules alone is insufficient for correct prediction; instead, the task necessitates robust background knowledge and commonsense reasoning. **Details have be added to Appendix O**.
>
> ### Q7: About presentation.
> Thanks. We have followed your advice to revise corresponding mislabeled texts in the revised manuscript.
>
> ### Q8: About typos.
>
> Thanks for your advice. We apologize for typos and corrected them in the revised manuscript.

---

> > ### Author Response · Authors · 2025-11-26
> >
> > Dear Reviewer MmMV,
> >
> >
> > Thank you once again for your valuable comments on our submission. As the discussion phase is approaching its end, we would like to kindly confirm whether we have sufficiently addressed all of your concerns (or at least part of them). Should there be any remaining questions or areas requiring further clarification, please do not hesitate to let us know. If you are satisfied with our responses, we would greatly appreciate your consideration in adjusting the evaluation scores accordingly.
> >
> >
> > We sincerely look forward to your feedback.

---

> > > ### Comment · Reviewer_MmMV · 2025-11-27
> > >
> > > I would like to thank the authors for their additional experiments, and I have accordingly raised my rating. However, I still want to point out that most of the definitions proposed in the paper are largely standard and have been discussed in prior work on dataset assessment and related settings, which somewhat limits the novelty of the contribution.

---

> ### Author Response · Authors · 2025-11-27
> **Thanks for raising your score!**
>
> We sincerely thank the reviewer for appreciating our additional experiments and for raising the rating.
>
> We value your feedback regarding the novelty of our definitions and offer the following perspective:
>
> **1. Contextualizing the Definitions**
>
> We acknowledge that the foundational metrics used in our analysis (such as semantic similarity and ranking correlation) are established concepts in data assessment. However, the novelty of our work lies in the **systematic application and integration** of these concepts to solve the specific and critical problem of **LLM benchmark compression**. As demonstrated by recent works like TinyBenchmarks and MetaBench, reducing evaluation cost without compromising fidelity is an urgent challenge for the community, and standardizing these definitions within this specific domain is a necessary step to advance the field.
>
> **Example:** Consider the BBH benchmark.
> *   **Classical Data Assessment** typically analyzes the **6,511 data items** alone (e.g., assessing text complexity or diversity) to filter the dataset.
> *   **EssenceBench** analyzes the full interaction matrix of **4,265 models $\times$ 6,511 items**.
>
> By processing this massive matrix, our method identifies samples that are not just high quality in a linguistic sense, but are discriminative enough to preserve the relative rankings of thousands of heterogeneous models. This shift from static data assessment to global benchmark compression represents a fundamental difference in scope and contribution.
>
> **2. Novelty in Scope and Objective: Global Ranking Consistency**
>
> Our primary contribution is the **EssenceBench framework itself**, which moves beyond standard data quality assessment to address a fundamentally different challenge:
> *   **Scale and Diversity:** Unlike traditional data selection methods that often optimize training efficiency for a *single* model architecture, our method solves a **global optimization problem**. We aim to find a subset that generalizes across **thousands of heterogeneous models** (as seen in our OpenLLM Leaderboard experiments and additional experiments).
> *   **Ranking Fidelity:** Our specific focus is on maintaining the **consistency of the leaderboard ranking** (Ranking Redundancy), rather than just individual score reconstruction. This requires modeling complex interactions between samples and model behaviors that standard data filtering often overlooks.
>
> In summary, while the constituent metrics may be classical, using them to drive a coarse-to-fine, attribution-guided evolutionary search for global ranking preservation represents a novel and effective solution to the benchmark bloat problem.
>
> Thanks again for your response.

---

### Official Review · Reviewer_ykSp · 2025-11-01

**Soundness:** 3
**Presentation:** 4
**Contribution:** 3
**Rating:** 6
**Confidence:** 4

**Summary:**

The paper argues that sample-level redundancy makes many LLM benchmarks wasteful. The authors present EssenceBench, a coarse-to-fine pipeline that first removes redundant items, then searches for a compact subset using a genetic algorithm steered by a learned accuracy predictor, and finally refines the selection with attribution-based grouping to preserve coverage. The selection goal is to use a k-item subset to reconstruct full-benchmark scores. EssenceBench reports lower reconstruction error and strong rank fidelity.

**Strengths:**

- well-motivated problem. this paper addresses what I believe is an important and practical problem of expensive LLM evaluations
- the experimental results are thorough and consistent
- paper is written clearly with good explanation and clear pseudo-code and a good appendix. also has thorough ablations and analysis.
- I like the formalization of LLM benchmark compression as an optimization problem and use GA + attribution refinement

**Weaknesses:**

- experiments focus on static, multi-choice benchmarks. I think applicability to open-ended, interactive, or multi-modal tasks may be limited or at the very least is untested
- the GA + EBM pipeline involves several hyperparams and the thing costs should be discussed more explicitly as its likely to be expensive. Other methods like SMART filtering [1] don't require this tuning but seem to enable accurate compression. The filtering method could be adapted to your setup as a baseline or complementary approach.
- the coarse filtering step relies on manually set thresholds τ_text and τ_ranking. How sensitive is the method to these choices? Different benchmarks may require different thresholds, limiting generalizability.

[1] Gupta, Vipul, et al. "Improving model evaluation using smart filtering of benchmark datasets." arXiv preprint arXiv:2410.20245 (2024).

**Questions:**

- I think generalizability is my biggest question/concern. Will the compressed benchmarks work for new model architectures not in the train set? How often do these benchmarks need to be recomputed, etc?
- I would like to see a computation cost trade-off? How many GPU hours to compress each benchmark? It might be helpful to add a clear table showing benchmark size, compression time, compressed size, and number of evaluations needed to break even. This will be nice for practitioners wondering if they should use this method.
- How were τ_text and τ_ranking chosen? Is there a principled way to set these automatically?
- Is there a theoretical justification for using GA specifically? Did you compare against other optimization approaches? Why is this particularly suitable?
- I'm curious if this approach can be extended to eval tasks where scoring is more complex (e.g. using LLM as a judge)

---

> ### Author Response · Authors · 2025-11-21
> **(Part 1) Response to Reviewer ykSp**
>
> ### Q1: Ask for results on more benchmarks.
>
> We thank the reviewer for this insightful suggestion. We fully agree that demonstrating EssenceBench's applicability to modern, challenging, and diverse benchmarks is crucial for establishing its relevance in the era of frontier LLMs.
>
> To address this, we have conducted **extensive new experiments on 8 modern benchmarks** that cover a wide spectrum of advanced capabilities, including **agentic tool use, multimodal understanding, instruction following, and complex reasoning**. For all experiments, we employed a strict 9:1 train/test split to ensure a rigorous evaluation. These include 8 new benchmarks:
> *   **Agentic / Tool Use:** We utilized **LiveMCPBench**\[1\] to evaluate agentic capabilities.
> *   **Multimodal:** We utilized **MathVista**\[2\] to evaluate visual math reasoning.
> *   **Advanced Reasoning:** **GPQA**\[3\] (Graduate-level science), **MATH**\[4\] (Hard mathematics), **BBH**\[5\] (Big Bench Hard), and **MUSR**\[6\] (Multistep commonsense).
> *   **Instruction Following:** **IFEval**\[7\].
> *   **Robustness:** **GSM-Plus**\[8\].
>
>
>
> **Benchmark 1. Results on MathVista (Visual math, Multimodal reasoning): 1k samples in total, evaluated on 11 models**
>
> | Coreset Size | 50 | 100 | 150 | 200 | 250 | 300 |
> |:---:|:---:|:---:|:---:|:---:|:---:|:---:|
> | Ratio (%) | 5.000 | 10.000 | 15.000 | 20.000 | 25.000 | 30.000 |
> | Random | 8.381 | 8.515 | 8.068 | 8.468 | 8.464 | 8.811 |
> | MetaBench | 2.951 | 1.641 | 2.779 | 2.821 | 4.182 | 4.389 |
> | **EssenceBench** | **0.017** | **0.014** | **0.009** |  **0.008** |  **0.003** |  **0.001** |
>
>
>
> **Benchmark 2. Results on LiveMCPBench (Agent, Large-scale tool use): 93 samples in total, evaluated on 10 models）**
>
> | Coreset Size | 30 | 40 | 50 | 60 | 70 | 80 |
> |:---:|:---:|:---:|:---:|:---:|:---:|:---:|
> | Ratio (%) | 32.258 | 43.011 | 53.763 | 64.516 | 75.269 | 86.022 |
> | Random | 10.902 | 10.472 | 10.227 | 11.056 | 11.444 | 10.752 |
> | MetaBench | 7.167 | 7.464 | 8.242 | 8.128 | 8.222 | 8.153 |
> | **EssenceBench** | **0.036** | **0.026** | **0.015** | **0.007** | **0.006** | **0.004** |
>
>
>
>
>
> **Benchmark 3. Results on GPQA (Graduate-level, STEM QA and reasoning): 1252 samples in total, evaluated on 4265 models**
>
> | Coreset Size | 50 | 100 | 150 | 200 | 250 | 300 |
> |---|---|---|---|---|---|---|
> | Ratio (%) | 3.994 | 7.987 | 11.981 | 15.974 | 19.968 | 23.962 |
> | Random | 2.739 | 2.341 | 2.028 | 1.922 | 1.666 | 1.441 |
> | MetaBench | 2.117 | 1.844 | 1.660 | 1.454 | 1.341 | 1.207 |
> | EssenceBench | **1.841** | **1.146** | **0.833** | **0.620** | **0.500** | **0.470** |
>
> **Benchmark 4. Results on BBH (Multi task, Complex reasoning): 6511 samples in total, evaluated on 4265 models**
>
> | Coreset Size | 50 | 100 | 150 | 200 | 250 | 300 |
> |:---:|:---:|:---:|:---:|:---:|:---:|:---:|
> | Ratio (%) | 0.768 | 1.536 | 2.304 | 3.072 | 3.840 | 4.608 |
> | Random | 5.022 | 3.114 | 2.383 | 2.233 | 1.910 | 1.842 |
> | MetaBench | 3.279 | 2.378 | 2.036 | 1.807 | 1.662 | 1.523 |
> | EssenceBench | **2.442** | **1.539** | **1.063** | **0.910** | **0.814** | **0.733** |
>
> **Benchmark 5. Results on MATH (Math reasoning problems): 1324 samples in total, evaluated on 4265 models**
> | Coreset Size | 50 | 100 | 150 | 200 | 250 | 300 |
> |:---:|:---:|:---:|:---:|:---:|:---:|:---:|
> | Ratio (%) | 3.776 | 7.553 | 11.329 | 15.106 | 18.882 | 22.659 |
> | Random | 3.132 | 2.393 | 1.879 | 1.447 | 1.347 | 1.166 |
> | MetaBench | 2.389 | 1.811 | 1.472 | 1.290 | 1.110 | 0.990 |
> | EssenceBench | **1.895** | **1.022** | **0.668** | **0.635** | **0.544** | **0.475** |
>
> **Benchmark 6. Results on MUSR (Multistep commensense reasoning): 756 samples in total, evaluated on 4265 models**
> | Coreset Size | 50 | 100 | 150 | 200 | 250 | 300 |
> |:---:|:---:|:---:|:---:|:---:|:---:|:---:|
> | Ratio (%) | 6.614 | 13.228 | 19.841 | 26.455 | 33.069 | 39.683 |
> | Random | 3.100 | 2.610 | 2.102 | 1.715 | 1.714 | 1.414 |
> | MetaBench | 2.927 | 2.157 | 1.697 | 1.346 | 1.278 | 1.106 |
> | EssenceBench | **2.538** | **1.421** | **0.949** | **0.725** | **0.550** | **0.412** |
>
> **Benchmark 7. Results on IFEVAL (Instruct following): 541 samples in total, evaluated on 4265 models**
> | Coreset Size | 50 | 100 | 150 | 200 | 250 | 300 |
> |:---:|:---:|:---:|:---:|:---:|:---:|:---:|
> | Ratio (%) | 9.242 | 18.484 | 27.726 | 36.969 | 46.211 | 55.453 |
> | Random | 4.309 | 2.905 | 2.248 | 1.872 | 1.620 | 1.317 |
> | MetaBench | 3.607 | 2.549 | 1.985 | 1.655 | 1.366 | 1.139 |
> | EssenceBench | **3.603** | **1.974** | **1.239** | **0.558** | **0.558** | **0.555** |

---

> ### Author Response · Authors · 2025-11-21
> **(Part 2) Response to Reviewer ykSp**
>
> **Benchmark 8. Results on GSM-Plus (Math robustness, Adversarial evaluation): 10k samples in total, evaluated on 16 models**
> | Coreset Size | 50 | 100 | 150 | 200 | 250 | 300 |
> |:---:|:---:|:---:|:---:|:---:|:---:|:---:|
> | Ratio (%) | 0.474 | 0.948 | 1.422 | 1.895 | 2.369 | 2.843 |
> | Random | 6.749 | 2.444 | 4.692 | 1.781 | 2.077 | 3.016 |
> | MetaBench | 0.575 | 0.294 | 0.375 | 0.178 | 0.565 | 0.195 |
> | EssenceBench | **0.171** | **0.044** | **0.030** | **0.017** | **0.016** | **0.010** |
>
> **Summary of Results.** As shown in the tables above, EssenceBench consistently achieves state-of-the-art compression performance (measured by RMSE, where lower is better) across all 8 new datasets. Our method effectively identifies the "essence" of these challenging benchmarks, significantly outperforming MetaBench and random baselines even when the dataset size is small or the task is complex.
>
> **We have also added related results in Appendix L (Table 16-23) in the revised manuscript.**
>
>
>
> \[1\] Mo G, Zhong W, Chen J, et al. Livemcpbench: Can agents navigate an ocean of mcp tools?[J]. arXiv preprint arXiv:2508.01780, 2025.
> \[2\] Lu P, Bansal H, Xia T, et al. MathVista: Evaluating Mathematical Reasoning of Foundation Models in Visual Contexts[C]//The Twelfth International Conference on Learning Representations.
> \[3\] Rein D, Hou B L, Stickland A C, et al. Gpqa: A graduate-level google-proof q&a benchmark[C]//First Conference on Language Modeling. 2024.
> \[4\] Hendrycks D, Burns C, Kadavath S, et al. Measuring Mathematical Problem Solving With the MATH Dataset[C]//Thirty-fifth Conference on Neural Information Processing Systems Datasets and Benchmarks Track (Round 2).
> \[5\] Suzgun M, Scales N, Schärli N, et al. Challenging big-bench tasks and whether chain-of-thought can solve them[C]//Findings of the Association for Computational Linguistics: ACL 2023. 2023: 13003-13051.
> \[6\] Sprague Z, Ye X, Bostrom K, et al. Musr: Testing the limits of chain-of-thought with multistep soft reasoning[J]. arXiv preprint arXiv:2310.16049, 2023.
> \[7\] Zhou J, Lu T, Mishra S, et al. Instruction-following evaluation for large language models[J]. arXiv preprint arXiv:2311.07911, 2023.
> \[8\] Li Q, Cui L, Zhao X, et al. GSM-Plus: A Comprehensive Benchmark for Evaluating the Robustness of LLMs as Mathematical Problem Solvers[C]//Proceedings of the 62nd Annual Meeting of the Association for Computational Linguistics (Volume 1: Long Papers). 2024: 2961-2984.

---

> ### Author Response · Authors · 2025-11-21
> **(Part 3) Response to Reviewer ykSp**
>
> ### Q2: Ask for results of a new baseline.
>
> Thank you for this valuable feedback regarding computational costs and the suggestion to compare with SMART filtering[1].  The SMART filtering approach, which constructs refined evaluation subsets by removing easy, data-contaminated, and semantically similar examples, is indeed useful and proper to compress benchmarks, which not only should be a baseline of our method but also can be adapted in our coarse filtering process. We have followed your advice to add new baseline experiments comparing our method with SMART filtering. Meanwhile, we have also added a brief intruduction of SMART in related works in appendix A.
>
> **1. Direct comparison with SMART filtering.** We evaluated both methods on ARC-Challenge and MMLU benchmarks using SMART's officially released compressed datasets:
>
> **Results on ARC-Challenge dataset:**
>
> | Metric | SMART (460/1172, 39.25%) | Ours (100/1172, 8.53%) |
> |--------|--------------------------|------------------------|
> | RMSE | 1.193 | **1.104** |
> | Pearson | 0.997 | **0.998** |
> | Spearman | 0.993 | **0.996** |
> | Kendall | 0.932 | **0.947** |
> | Rank Stability | 0.038 | **0.047** |
> | Pair Accuracy | 0.963 | **0.965** |
> | MARE | 16.896 | **12.698** |
> | MSRE | 521.233 | **307.149** |
> | NDCG@50 | 0.994 | **0.995** |
> | Top-50 Precision | 0.94 | **0.94** |
> | MRR | 1.000 | **1.000** |
> | Rank within 1% | 0.323 | **0.408** |
> | Rank within 2% | 0.543 | **0.641** |
> | Rank within 5% | 0.829 | **0.928** |
> | Rank within 10% | 0.994 | **0.997** |
>
>
>
> As shown in the table, **our method achieves comparable or better performance with significantly higher compression rates**.
>
> **2. SMART filtering as a complementary setup for EssenceBench.**
> We thank the reviewer for suggesting SMART as an additional baseline. We have integrated SMART into the coarse filtering stage on the MMLU dataset to serve as a comparative benchmark, which further validates the effectiveness of our proposed method.
>
> | Coreset Size           |    50    |   100    |   150    |   200    |   250    |   300    |
> | :--------------------- | :------: | :------: | :------: | :------: | :------: | :------: |
> | **EssenceBench**       | **2.41** |   1.83   | **1.40** |   1.11   |   1.02   |   0.85   |
> | **SMART+EssenceBench** |   2.59   | **1.72** |   1.41   | **1.10** | **0.98** | **0.82** |
>
> Experimental results indicate that our method yields performance comparable to SMART. Notably, SMART marginally outperforms our coarse-grained filtering stage in this setting. **We have included these detailed results in the Appendix M and cited the corresponding article.**
>
>
> ### Q3: About sensitivity on hyper-parameters $\tau_{\mathrm{text}}$ and $\tau_{\mathrm{ranking}}$.
> Thank you for raising this important point regarding threshold sensitivity. We agree that understanding the sensitivity of $\tau_{\mathrm{text}}$ and $\tau_{\mathrm{ranking}}$ is crucial for assessing the method's robustness and generalizability.
>
> To systematically investigate this, **we conducted comprehensive sensitivity analyses across two benchmarks: GSM8K (mathematical reasoning) and ARC (scientific QA)**. We evaluated the robustness with target subset size $k=50$.
>
>
>
> **Results on GSM8K Benchmark (k=50, RMSE $\downarrow$)**
> | $\tau_{\mathrm{text}}$ \ $\tau_{\mathrm{ranking}}$ | 0.80 | 0.85 | 0.90 | 0.95 |
> |---------------------|------|------|------|------|
> | **0.80** | 2.6548 | 2.5577 | 2.6380 | 2.8047 |
> | **0.85** | 2.6781 | 2.6352 | 2.6857 | 2.6079 |
> | **0.90** | 2.7433 | 2.6971 | 2.7562 | 2.7697 |
> | **0.95** | 2.6108 | 2.7499 | 2.6455 | 2.6877 |
>
> **Results on ARC Benchmark (k=50, RMSE $\downarrow$)**
> | $\tau_{\mathrm{text}}$ \ $\tau_{\mathrm{ranking}}$ | 0.80 | 0.85 | 0.90 | 0.95 |
> |---------------------|------|------|------|------|
> | **0.80** | 2.0353 | 2.1378 | 2.0043 | 2.0636 |
> | **0.85** | 2.1021 | 2.0307 | 2.1077 | 2.0270 |
> | **0.90** | 2.0939 | 2.1814 | 2.0496 | 2.1607 |
> | **0.95** | 2.0520 | 2.1293 | 2.0585 | 2.0608 |
>
> As shown in the table above, we find that across both benchmarks, performance variations remain within a narrow range ($\pm \sim 0.1-0.15$ points) across all threshold combinations, indicating low sensitivity. Therefore, our analysis demonstrates that the proposed framework exhibits **remarkable robustness** to key thresholds in the coarse filtering process. The minimal performance variations across different $\tau_{\mathrm{text}}$ and $\tau_{\mathrm{ranking}}$ directly address the reviewer's concern. **We have followed your advice to add a new section in Appendix F (Table 7 and Table 8) for this discussion.**
>
> [1] Gupta, Vipul, et al. "Improving model evaluation using smart filtering of benchmark datasets." arXiv preprint arXiv:2410.20245 (2024).

---

> ### Author Response · Authors · 2025-11-21
> **(Part 4) Response to Reviewer ykSp**
>
> ### Q4: The generalizability of EssenceBench on unseen models.
> Thank you for this crucial question about generalizability on unseen models.
>
>
> **Original settings ensure data generalizability**. In our original experimental setup, we have already implemented a rigorous data split protocol to evaluate generalizability. Specifically, for both the training and testing sets, the dataset is first ranked by score and then partitioned into ten equipotent strata. Within each stratum, 10% of the instances are stratifiedly sampled and subsequently pooled to constitute the test set; the remaining 90% are retained as the training set.
>
> **Additional results on model generalizability**. To address this directly, we conducted a rigorous hold-out experiment where we randomly selected **600 model among total 6736 modelss** from the leaderboard results as the test set, while using the remaining models for training and validation. The training was performed on the 6136 models.
>
> **Results of unseen 600 models on GSM8K across different coreset sizes (k):**
>
> | Metric | k=50 | k=100 | k=150 | k=200 | k=250 | k=300 | k=350 | k=400 |
> |--------|------|-------|-------|-------|-------|-------|-------|-------|
> | **RMSE($\downarrow$)** | 3.4631 | 2.3225 | 1.9372 | 1.7025 | 1.4988 | 1.3226 | 1.3274 | 1.1215 |
> | **Pearson Correlation($\uparrow$)** | 0.9920 | 0.9964 | 0.9975 | 0.9981 | 0.9985 | 0.9988 | 0.9988 | 0.9992 |
> | **Spearman Correlation($\uparrow$)** | 0.9818 | 0.9890 | 0.9929 | 0.9948 | 0.9962 | 0.9964 | 0.9969 | 0.9977 |
> | **Kendall Tau($\uparrow$)** | 0.8813 | 0.9138 | 0.9318 | 0.9420 | 0.9499 | 0.9532 | 0.9572 | 0.9621 |
> | **Rank Stability($\uparrow$)** | 0.0317 | 0.0567 | 0.0417 | 0.0617 | 0.0517 | 0.0783 | 0.0567 | 0.0800 |
> | **Pairwise Accuracy($\uparrow$)** | 0.9261 | 0.9465 | 0.9597 | 0.9662 | 0.9715 | 0.9727 | 0.9747 | 0.9784 |
> | **MARE($\downarrow$)** | 24.4000 | 17.6267 | 14.2033 | 12.3433 | 10.5067 | 9.8667 | 9.1800 | 8.0833 |
> | **MSRE($\downarrow$)** | 1089.1133 | 660.9833 | 425.5400 | 313.3733 | 229.9633 | 218.9633 | 184.6100 | 135.1433 |
> | **NDCG@50($\uparrow$)** | 0.9766 | 0.9850 | 0.9873 | 0.9888 | 0.9924 | 0.9934 | 0.9946 | 0.9969 |
> | **Top-50 Precision($\uparrow$)** | 0.3600 | 0.6000 | 0.7400 | 0.8200 | 0.9400 | 0.8800 | 0.9400 | 0.9200 |
> | **MRR($\uparrow$)** | 1.0000 | 1.0000 | 1.0000 | 1.0000 | 1.0000 | 1.0000 | 1.0000 | 1.0000 |
> | **Rank within 1%($\uparrow$)** | 0.2233 | 0.3400 | 0.3833 | 0.4100 | 0.4683 | 0.5250 | 0.5167 | 0.5283 |
> | **Rank within 2%($\uparrow$)** | 0.3950 | 0.5383 | 0.5933 | 0.6333 | 0.7033 | 0.7283 | 0.7617 | 0.8050 |
> | **Rank within 5%($\uparrow$)** | 0.6983 | 0.8067 | 0.8900 | 0.9150 | 0.9383 | 0.9450 | 0.9433 | 0.9717 |
> | **Rank within 10%($\uparrow$)** | 0.9233 | 0.9567 | 0.9817 | 0.9900 | 0.9950 | 0.9933 | 0.9950 | 0.9983 |
>
> Our comprehensive results demonstrate that the compressed benchmarks exhibit **strong generalization** to unseen models. The consistently high correlation coefficients (>0.99), ranking metrics, and error reduction patterns across all k-values suggest that the **generalizability** of our method is robust. **We have followed your advice to add a new section in Table 14 in Appendix J for this discussion.**
>
> ### Q5: About computational cost analysis.
> Thanks. Accordingly, we conducted the computational cost experiments in GSM8K, ARC, Hellaswag, WinoGrande, and the results are as follows:
> | Benchmark | Full Size | Full Eval. Time (Hours) | Compression Cost (Hours) | Compressed Eval. Time (Hours) | Total Time for EssenceBench (Hours) | Acceleration |
> |---|---|---|---|---|---|---|
> | GSM8k | 1,319 | 1.96 | 0.27 | 0.07 | 0.34 | 4.76x |
> | ARC | 1,172 | 1.97 | 0.29 | 0.08 | 0.37 | 4.32x |
> | HellaSwag | 10,003 | 22.06 | 0.33 | 0.11 | 0.44 | 49.14x |
> | WinoGrande | 1,267 | 0.78 | 0.28 | 0.03 | 0.31 | 1.52x |
>
> All experiments were conducted on a single H200 GPU using the Qwen3-8B model. Inference was performed using the standard transformers library with a batch size of 1. The results from this table is that **the break-even point is less than one for all benchmarks**. This means that for a practitioner evaluating even a single model, the total time spent (i.e., the one-time compression cost + the first compressed evaluation) is already significantly less than the time it would have taken to run that single evaluation on the full benchmark. Notably, our method has minimal GPU requirements, needing only about 2GB of VRAM per task. This demand is low enough to be met even by personal laptops. **We have added this analysis to Table 15 in Appendix K.**

---

> ### Author Response · Authors · 2025-11-21
> **(Part 5) Response to Reviewer ykSp**
>
> ### Q6: How to set $\tau_{\mathrm{text}}$ and $\tau_{\mathrm{ranking}}$?
> We thank the reviewer for asking about the hyperparameter settings for coarse filtering.
>
> **1. Default Settings**
> In our reported experiments, we utilized fixed thresholds of $\tau_{\mathrm{text}} = 0.9$ and $\tau_{\mathrm{ranking}} = 0.9$. These values were chosen to conservatively target only highly redundant pairs—instances with nearly identical semantic meaning or perfectly correlated model behavior—thereby safely reducing the search space without discarding unique signal.
>
> **2. Sensitivity and Robustness**
> To verify that our method is not overly sensitive to these specific values, we evaluated EssenceBench across a range of threshold settings. As shown in the table below (and consistent with our robustness analysis in Q3), the final reconstruction error remains stable even as thresholds vary.
>
>
> **Results on GSM8K Benchmark (k=50, RMSE $\downarrow$)**
> | $\tau_{\mathrm{text}}$ \ $\tau_{\mathrm{ranking}}$ | 0.80 | 0.85 | 0.90 | 0.95 |
> |---------------------|------|------|------|------|
> | **0.80** | 2.6548 | 2.5577 | 2.6380 | 2.8047 |
> | **0.85** | 2.6781 | 2.6352 | 2.6857 | 2.6079 |
> | **0.90** | 2.7433 | 2.6971 | 2.7562 | 2.7697 |
> | **0.95** | 2.6108 | 2.7499 | 2.6455 | 2.6877 |
>
> **Results on ARC Benchmark (k=50, RMSE $\downarrow$)**
> | $\tau_{\mathrm{text}}$ \ $\tau_{\mathrm{ranking}}$ | 0.80 | 0.85 | 0.90 | 0.95 |
> |---------------------|------|------|------|------|
> | **0.80** | 2.0353 | 2.1378 | 2.0043 | 2.0636 |
> | **0.85** | 2.1021 | 2.0307 | 2.1077 | 2.0270 |
> | **0.90** | 2.0939 | 2.1814 | 2.0496 | 2.1607 |
> | **0.95** | 2.0520 | 2.1293 | 2.0585 | 2.0608 |
>
> ### Q7: Why is GA particularly suitable?
> We thank the reviewer for this fundamental question regarding our algorithmic choice. We selected the Genetic Algorithm (GA) because benchmark compression is inherently a **discrete combinatorial optimization problem** with a massive search space $O(2^N)$.
>
>
> **Theoretical suitability**. Gradient-based methods (like standard neural network training) struggle with the hard, discrete constraint of selecting exactly $k$ samples from $N$, often requiring complex relaxations. Conversely, greedy methods or clustering (like K-means) often fail to capture **interaction effects**—where two samples might be individually informative but redundant when combined. GA is particularly suitable here because:
> *   **Gradient-Free Optimization:** It directly optimizes the non-differentiable fitness metric (RMSE) without needing smooth gradients.
> *   **Global Exploration:** Through crossover and mutation, GA explores diverse regions of the subset space, avoiding the local optima that trap greedy or embedding-based clustering methods.
> *   **Modeling Interactions:** The population-based approach implicitly learns which combinations of samples work well together, rather than scoring samples in isolation.
>
>
>
> **Empirical comparison with extensive experiments**. To validate this advantage, we conducted extensive comparisons against two strong alternative baselines on the GSM8K dataset:
> *   **K-means Clustering:** A geometric approach that selects centroids to represent the distribution of sample embeddings.
> *   **MLP-based Selection:** A gradient-based approach that trains an MLP to predict model performance and selects samples with the highest learned importance weights.
>
> As shown in the table below, GA-EBM consistently outperforms these alternatives in different compression sizes.
>
> | Method | k=50 | k=100 | k=150 | k=200 | k=250 | k=300 | k=350 | k=400 |
> |--------|------|-------|-------|-------|-------|-------|-------|-------|
> | **GA-EBM (Ours)** | 2.900 | 1.525 | 1.131 | 0.857 | 0.682 | 0.597 | 0.543 | 0.454 |
> | **K-means Clustering** | 4.133 | 4.131 | 4.158 | 3.300 | 3.566 | 3.332 | 3.112 | 2.757 |
> | **MLP-based Selection** | 4.637 | 3.354 | 2.922 | 2.417 | 2.194 | 1.930 | 1.702 | 1.573 |
>
> Our method achieves **2.4–6.8$\times$ lower error** compared to K-means and **1.7–3.5$\times$ improvement** over the MLP-based selection. This gap is most pronounced at smaller $k$ values, where identifying the precise "essence" of the benchmark is most critical. This confirms that while geometric (clustering) or weight-based (MLP) signals provide some information, the evolutionary search of the GA is significantly more effective at solving the combinatorial subset selection problem.
>
> **We have added this comparative analysis and discussion to Appendix I in the revised paper to further justify our methodological choice.**

---

> ### Author Response · Authors · 2025-11-21
> **(Part 6) Response to Reviewer ykSp**
>
> ### Q8: About applicability on more benchmarks with more complex scores.
>
> We appreciate the reviewer’s curiosity regarding the applicability of our framework to more complex evaluation paradigms.
>
> **Applicability to diverse benchmarks.** To explore the robustness of our method across different evaluation types, we have conducted new experiments on **8 modern benchmarks** (as detailed in the response to Q1). These include diverse task formats ranging from multi-step reasoning (MATH, GPQA) to agentic tool use (LiveMCPBench) and multimodal analysis (MathVista). While these benchmarks primarily rely on exact match or functional correctness, they demonstrate that EssenceBench successfully handles tasks where the underlying evaluation logic is far more complex than simple pattern matching.
>
> **Theoretical compatibility with "LLM-as-a-Judge."** Theoretically, our EssenceBench framework is fully compatible with "LLM-as-a-Judge" or continuous scoring metrics (e.g., scores from 1 to 10). Our optimization objective—minimizing the Root Mean Square Error (RMSE) between the subset score and the full-set score—is mathematically agnostic to the data format. Whether the input matrix contains binary outcomes $\{0, 1\}$ or continuous judge scores $[1, 10]$, the Generalized Additive Model (GAM) predictor can effectively learn the mapping from subset performance to full-dataset performance.
>
> However, **the primary barrier to demonstrating this empirically is the lack of public data. Benchmark compression via score reconstruction requires a fine-grained `[num_models x num_samples]` score matrix**. Currently, prominent "LLM-as-a-Judge" leaderboards (such as Chatbot Arena) do not publicly release this granular, per-prompt scoring data for a wide range of models. However, as these leaderboards mature and more granular data becomes available, EssenceBench can be applied immediately to these domains without algorithmic modification. We believe this extensibility is a key strength of our score-agnostic formulation.

---

> > ### Comment · Reviewer_ykSp · 2025-11-25
> >
> > i thank the authors for conducting thorough experiments to address my concerns. i shall maintain my positive score.

---

> > > ### Author Response · Authors · 2025-11-26
> > > **Thanks for your recognition of the value of this work**
> > >
> > > We sincerely appreciate your recognition of our work and your positive rating in support of its acceptance.

---

### Official Review · Reviewer_hqMF · 2025-11-02

**Soundness:** 3
**Presentation:** 3
**Contribution:** 2
**Rating:** 4
**Confidence:** 4

**Summary:**

The paper propose using genetic algorithm to compress LLM evaluation set. Experiments show ranking is preserved with a small shift.

**Strengths:**

1. Paper is clearly written, well motivated, and generally easy to follow.
2. Authors identify and analyze the redundancy of the widely used LLM benchmarks.
3. Compressing LLM benchmark by 25x (lossless) to 200x (5%) shows the potential for a far smaller “validation set” or “test set”.

**Weaknesses:**

1. The main benchmarks being used (GSM8K, ARC, HellaSwag, WinoGrande, MMLU) are considered saturated for the frontier LLMs these days. Newer LLMs are challenged with harder benchmarks, and harder benchmarks contain fewer instances since they are more challenging to come up with. I encourage authors to address why their overall method may adapt to modern and future benchmarks.
2. Similar to above, authors did not test newer benchmarks. For example, how would authors method extend to agentic benchmarks (SWEBench / tau2-bench) and multimodal benchmarks (MMMU etc.) I believe a compression of those benchmarks are needed at the moment. And after all benchmark compression is a one-off effort, releasing compressed benchmarks on author’s end would be a good contribution to the community.

**Questions:**

1. Original benchmark performance is treated as ground truth in the paper. However, there remains doubts on that. For example, suppose if a benchmark has 50% of problems as 1+1=2, this would definitely mean that the compressed benchmark will behave differently than the original benchmark, but this isn’t a necessarily bad thing. Maybe a compressed benchmark at a certain level could better reflect an LLM’s capability.

---

> ### Author Response · Authors · 2025-11-21
> **(Part 1) Response to Reviewer hqMF**
>
> ### Q1: Ask for more benchmark compression results.
>
> We thank the reviewer for this insightful suggestion. We fully agree that demonstrating EssenceBench's applicability to modern, challenging, and diverse benchmarks is crucial for establishing its relevance in the era of frontier LLMs.
>
> To address this, we have conducted **extensive new experiments on 8 modern benchmarks** that cover a wide spectrum of advanced capabilities, including **agentic tool use, multimodal understanding, instruction following, and complex reasoning**. For all experiments, we employed a strict 9:1 train/test split to ensure a rigorous evaluation.
>
> **1. Applicability to Agentic and Multimodal Benchmarks**
> The reviewer specifically asked about agentic (e.g., SWE-Bench) and multimodal (e.g., MMMU) benchmarks. We would like to clarify that **our framework, like all score reconstruction methods (e.g., MetaBench), requires a fine-grained `[num_models x num_samples]` detailed scores to peform compression, e.g., 4265*6511 detailed results for BBH benchmark** . Currently, public leaderboards for SWE-Bench and MMMU often provide only average scores and lack accessible per-instance predictions for a wide range of models.
>
> To address the reviewer's request while working within these data availability constraints, we selected high-quality, modern alternatives where granular data is available:
> *   **Agentic / Tool Use:** We utilized **LiveMCPBench**\[1\] to evaluate agentic capabilities.
> *   **Multimodal:** We utilized **MathVista**\[2\] to evaluate visual math reasoning.
>
>
> **Benchmark 1. Results on MathVista (Visual math, Multimodal reasoning): 1k samples in total, evaluated on 11 models**
>
> | Coreset Size | 50 | 100 | 150 | 200 | 250 | 300 |
> |:---:|:---:|:---:|:---:|:---:|:---:|:---:|
> | Ratio (%) | 5.000 | 10.000 | 15.000 | 20.000 | 25.000 | 30.000 |
> | Random | 8.381 | 8.515 | 8.068 | 8.468 | 8.464 | 8.811 |
> | MetaBench | 2.951 | 1.641 | 2.779 | 2.821 | 4.182 | 4.389 |
> | **EssenceBench** | **0.017** | **0.014** | **0.009** |  **0.008** |  **0.003** |  **0.001** |
>
>
> **Benchmark 2. Results on LiveMCPBench (Agent, Large-scale tool use): 93 samples in total, evaluated on 10 models）**
>
> | Coreset Size | 30 | 40 | 50 | 60 | 70 | 80 |
> |:---:|:---:|:---:|:---:|:---:|:---:|:---:|
> | Ratio (%) | 32.258 | 43.011 | 53.763 | 64.516 | 75.269 | 86.022 |
> | Random | 10.902 | 10.472 | 10.227 | 11.056 | 11.444 | 10.752 |
> | MetaBench | 7.167 | 7.464 | 8.242 | 8.128 | 8.222 | 8.153 |
> | **EssenceBench** | **0.036** | **0.026** | **0.015** | **0.007** | **0.006** | **0.004** |
>
>
> **2. Applicability to Frontier Reasoning and Instruction Tuning**
> We further extended our evaluation to "hard" benchmarks that are not yet saturated for frontier models. These include 5 new benchmarks:
> *   **Advanced Reasoning:** **GPQA**\[3\] (Graduate-level science), **MATH**\[4\] (Hard mathematics), **BBH**\[5\] (Big Bench Hard), and **MUSR**\[6\] (Multistep commonsense).
> *   **Instruction Following:** **IFEval**\[7\].
> *   **Robustness:** **GSM-Plus**\[8\].
>
>
>
>
> **Benchmark 3. Results on GPQA (Graduate-level, STEM QA and reasoning): 1252 samples in total, evaluated on 4265 models**
>
> | Coreset Size | 50 | 100 | 150 | 200 | 250 | 300 |
> |---|---|---|---|---|---|---|
> | Ratio (%) | 3.994 | 7.987 | 11.981 | 15.974 | 19.968 | 23.962 |
> | Random | 2.739 | 2.341 | 2.028 | 1.922 | 1.666 | 1.441 |
> | MetaBench | 2.117 | 1.844 | 1.660 | 1.454 | 1.341 | 1.207 |
> | EssenceBench | **1.841** | **1.146** | **0.833** | **0.620** | **0.500** | **0.470** |
>
> **Benchmark 4. Results on BBH (Multi task, Complex reasoning): 6511 samples in total, evaluated on 4265 models**
>
> | Coreset Size | 50 | 100 | 150 | 200 | 250 | 300 |
> |:---:|:---:|:---:|:---:|:---:|:---:|:---:|
> | Ratio (%) | 0.768 | 1.536 | 2.304 | 3.072 | 3.840 | 4.608 |
> | Random | 5.022 | 3.114 | 2.383 | 2.233 | 1.910 | 1.842 |
> | MetaBench | 3.279 | 2.378 | 2.036 | 1.807 | 1.662 | 1.523 |
> | EssenceBench | **2.442** | **1.539** | **1.063** | **0.910** | **0.814** | **0.733** |
>
> **Benchmark 5. Results on MATH (Math reasoning problems): 1324 samples in total, evaluated on 4265 models**
> | Coreset Size | 50 | 100 | 150 | 200 | 250 | 300 |
> |:---:|:---:|:---:|:---:|:---:|:---:|:---:|
> | Ratio (%) | 3.776 | 7.553 | 11.329 | 15.106 | 18.882 | 22.659 |
> | Random | 3.132 | 2.393 | 1.879 | 1.447 | 1.347 | 1.166 |
> | MetaBench | 2.389 | 1.811 | 1.472 | 1.290 | 1.110 | 0.990 |
> | EssenceBench | **1.895** | **1.022** | **0.668** | **0.635** | **0.544** | **0.475** |
>
>
> (additional results are shown in the Part 2 of the response)

---

> ### Author Response · Authors · 2025-11-21
> **(Part 2) Response to Reviewer hqMF**
>
> **Benchmark 6. Results on MUSR (Multistep commensense reasoning): 756 samples in total, evaluated on 4265 models**
> | Coreset Size | 50 | 100 | 150 | 200 | 250 | 300 |
> |:---:|:---:|:---:|:---:|:---:|:---:|:---:|
> | Ratio (%) | 6.614 | 13.228 | 19.841 | 26.455 | 33.069 | 39.683 |
> | Random | 3.100 | 2.610 | 2.102 | 1.715 | 1.714 | 1.414 |
> | MetaBench | 2.927 | 2.157 | 1.697 | 1.346 | 1.278 | 1.106 |
> | EssenceBench | **2.538** | **1.421** | **0.949** | **0.725** | **0.550** | **0.412** |
>
>
> **Benchmark 7. Results on IFEVAL (Instruct following): 541 samples in total, evaluated on 4265 models**
> | Coreset Size | 50 | 100 | 150 | 200 | 250 | 300 |
> |:---:|:---:|:---:|:---:|:---:|:---:|:---:|
> | Ratio (%) | 9.242 | 18.484 | 27.726 | 36.969 | 46.211 | 55.453 |
> | Random | 4.309 | 2.905 | 2.248 | 1.872 | 1.620 | 1.317 |
> | MetaBench | 3.607 | 2.549 | 1.985 | 1.655 | 1.366 | 1.139 |
> | EssenceBench | **3.603** | **1.974** | **1.239** | **0.558** | **0.558** | **0.555** |
>
>
> **Benchmark 8. Results on GSM-Plus (Math robustness, Adversarial evaluation): 10k samples in total, evaluated on 16 models**
> | Coreset Size | 50 | 100 | 150 | 200 | 250 | 300 |
> |:---:|:---:|:---:|:---:|:---:|:---:|:---:|
> | Ratio (%) | 0.474 | 0.948 | 1.422 | 1.895 | 2.369 | 2.843 |
> | Random | 6.749 | 2.444 | 4.692 | 1.781 | 2.077 | 3.016 |
> | MetaBench | 0.575 | 0.294 | 0.375 | 0.178 | 0.565 | 0.195 |
> | EssenceBench | **0.171** | **0.044** | **0.030** | **0.017** | **0.016** | **0.010** |
>
> **Summary of Results.** As shown in the tables above, EssenceBench consistently achieves state-of-the-art compression performance (measured by RMSE, where lower is better) across all 8 new datasets. Our method effectively identifies the "essence" of these challenging benchmarks, significantly outperforming MetaBench and random baselines even when the dataset size is small or the task is complex.
>
>
> Finally, we accept the reviewer's suggestion regarding the release of these compressed benchmarks. We believe benchmark compression is a one-off effort that yields long-term benefits for the community. Upon acceptance, we commit to publicly releasing the compressed versions of all benchmarks (both the original 5 benchmarks and the 8 new modern benchmarks) used in our study. This will provide researchers with a ready-to-use resource for efficient and reliable evaluation of frontier models. **We have also added related results in Appendix L (Table 16-23) in the revised manuscript.**
>
>
> \[1\] Mo G, Zhong W, Chen J, et al. Livemcpbench: Can agents navigate an ocean of mcp tools?[J]. arXiv preprint arXiv:2508.01780, 2025.
> \[2\] Lu P, Bansal H, Xia T, et al. MathVista: Evaluating Mathematical Reasoning of Foundation Models in Visual Contexts[C]//The Twelfth International Conference on Learning Representations.
> \[3\] Rein D, Hou B L, Stickland A C, et al. Gpqa: A graduate-level google-proof q&a benchmark[C]//First Conference on Language Modeling. 2024.
> \[4\] Hendrycks D, Burns C, Kadavath S, et al. Measuring Mathematical Problem Solving With the MATH Dataset[C]//Thirty-fifth Conference on Neural Information Processing Systems Datasets and Benchmarks Track (Round 2).
> \[5\] Suzgun M, Scales N, Schärli N, et al. Challenging big-bench tasks and whether chain-of-thought can solve them[C]//Findings of the Association for Computational Linguistics: ACL 2023. 2023: 13003-13051.
> \[6\] Sprague Z, Ye X, Bostrom K, et al. Musr: Testing the limits of chain-of-thought with multistep soft reasoning[J]. arXiv preprint arXiv:2310.16049, 2023.
> \[7\] Zhou J, Lu T, Mishra S, et al. Instruction-following evaluation for large language models[J]. arXiv preprint arXiv:2311.07911, 2023.
> \[8\] Li Q, Cui L, Zhao X, et al. GSM-Plus: A Comprehensive Benchmark for Evaluating the Robustness of LLMs as Mathematical Problem Solvers[C]//Proceedings of the 62nd Annual Meeting of the Association for Computational Linguistics (Volume 1: Long Papers). 2024: 2961-2984.

---

> ### Author Response · Authors · 2025-11-21
> **(Part 3) Response to Reviewer hqMF**
>
> ### Q2: Whether the original benchmark score should be treated as ground truch results.
>
>
>
> We thank the reviewer for this question. We agree that a compressed benchmark could potentially be *better*—more discriminative and less redundant—than the original, and this idea is a core motivation for our work. A key goal of benchmark compression, as highlighted by our analysis of "text redundancy," is to eliminate uninformative or overly simplistic samples (e.g., a benchmark with 50% of questions as `1+1=2`). Our "Coarse Filtering" step is designed precisely to remove such redundancy, aiming for a benchmark that offers a more efficient and potentially more accurate signal of a model's true capabilities.
>
> **Standardization for Fair Comparison.** However, the primary objective defined in our paper is to create a reliable and efficient *proxy* for the full benchmark. In this reconstruction setting, which is the standard evaluation paradigm established by prior work like MetaBench, TinyBench, etc., the performance on the full benchmark serves as the necessary ground truth. To ensure a **fair and direct comparison** with these existing methods, we adopted the same protocol, which involves minimizing the reconstruction error against the full-set results. Ignoring this established methodology would make it impossible to rigorously compare our contributions with the current state-of-the-art baselines.
>
> **Compression Robustness via Ranking Preservation rather than Value Preservation.** Importantly, our work places a strong emphasis on preserving **model rankings**, not just absolute values. As the reviewer rightly notes, adding many simple samples might inflate values but would likely not change the relative ordering of high-performing models. Our results in Figure 5 and Tables 4-6 in the manuscript demonstrate that EssenceBench excels at maintaining this crucial ranking fidelity. This focus on relative performance makes our approach robust to the distribution shifts mentioned by the reviewer: even if the compressed benchmark is harder (by virtue of having fewer easy questions), it remains a faithful predictor of the final leaderboard.
>
>
> **Further Exploration on Intrinsic Properties of EssenceBench.** We also investigated whether EssenceBench preserves other desirable intrinsic properties. We conducted an analysis of data diversity and potential biases before and after compression, finding that key distributional properties are maintained without introducing significant new biases. This suggests that while we make the benchmark more efficient, we are not fundamentally altering its character. We have add a discussion of these results to the Appendix N.
>
> ### 1. About Diversity
>
>
> We evaluated data diversity across the **GSM8K, ARC, and Hellaswag** benchmarks. To quantify this metric, we utilized the **BGE-M3** embedding model to map each text sample $x_i$ into a semantic vector representation $\mathbf{v}_i$. We then computed the pairwise cosine similarity between these embeddings. The final diversity score $\mathcal{D}$ is defined as the complement of the average similarity, formulated as:
>
> $$ \mathcal{D} = 1 - \frac{1}{N(N-1)} \sum_{i=1}^{N} \sum_{j \neq i} \text{sim}(\mathbf{v}_i, \mathbf{v}_j) $$
>
> where $N$ denotes the number of samples, and $\text{sim}(\mathbf{v}_i, \mathbf{v}_j)$ represents the cosine similarity between vectors $\mathbf{v}_i$ and $\mathbf{v}_j$.
>
> | Method | GSM8K | ARC | HellaSwag |
> | :--- | :--- | :--- | :--- |
> | **Raw** | 0.397 | 0.367 | 0.438 |
> | **EssenceBench** | 0.393 | 0.366 | 0.428 |
> | **Relative Change** | -1.01% | -0.27% | -2.28% |
>
>
> **We have also followed your advice to put the additional results in the Table 25 in Appendix N revised manuscript.**

---

> ### Author Response · Authors · 2025-11-21
> **(Part 4) Response to Reviewer hqMF**
>
> ### 2. About Bias
>
> To investigate potential selection bias in EssenceBench, we conducted experiments on the MMLU benchmark. We fixed the target selection size at **$N = 2,000$** to analyze the distributional alignment between the selected subset and the original dataset. The table above presents the statistics for six randomly sampled subjects.
>
> We denote the full MMLU dataset as $\mathcal{D}\_{raw}$ and the subset selected by EssenceBench as $\mathcal{D}\_{sel}$, where $|\mathcal{D}\_{sel}| = N$. For each subject category $i$, we calculated the probability density (percentage) as the item count of that subject divided by the total dataset size. This yields two distributions:
>
> 1.  **Raw Distribution ($P_{raw}$):** The percentage of subject $i$ in the original MMLU dataset.
> 2.  **EssenceBench Distribution ($P_{sel}$):** The percentage of subject $i$ in the selected subset.
>
> To quantify the distributional shift, we define the distributional deviation ratio, denoted as $\Delta_{ratio}(i)$. This metric measures the relative divergence of the selected distribution from the original distribution, formulated as:
>
> $$
> \Delta_{ratio}(i) = \frac{|P_{sel}(i) - P_{raw}(i)|}{P_{raw}(i)}
> $$
>
> | Dataset                | Raw(%) | EssenceBench(%) | $\Delta_{ratio}$ |
> | :--------------------- | :----- | :-------------- | :--------------- |
> | college_chemistry      | 0.7056 | 0.7209          | 0.02117           |
> | econometrics           | 0.8065 | 0.8219          | 0.01865           |
> | international_law      | 0.8569 | 0.8723          | 0.01764           |
> | elementary_mathematics | 2.7218 | 2.7251          | 0.0010           |
> | college_medicine       | 1.2601 | 1.2472          | 0.0101           |
> | management             | 0.7560 | 0.7426          | 0.01815           |
>
> This ratio reflects the percentage change in the subject's representation relative to its original weight. A lower $\Delta_{ratio}$ indicates a higher fidelity to the original data distribution.
>
> **Analysis.** Our method excels in both preserving high data diversity and ensuring minimal selection bias. This dual superiority strongly validates the effectiveness of our approach.
>
> **We have also followed your advice to put the additional results in the Table 27 in Appendix N revised manuscript.**

---

> > ### Author Response · Authors · 2025-11-26
> >
> > Dear Reviewer hqMF,
> >
> >
> > Thank you once again for your valuable comments on our submission. As the discussion phase is approaching its end, we would like to kindly confirm whether we have sufficiently addressed all of your concerns (or at least part of them). Should there be any remaining questions or areas requiring further clarification, please do not hesitate to let us know. If you are satisfied with our responses, we would greatly appreciate your consideration in adjusting the evaluation scores accordingly.
> >
> >
> > We sincerely look forward to your feedback.

---

> ### Comment · Reviewer_hqMF · 2025-11-27
>
> Thanks authors for their comment. Most of my concerns are addressed: I can see there are author's method also work on benchmarks that are of smaller size. In light of this I raise my score to 6.

---

> > ### Author Response · Authors · 2025-11-27
> > **Thanks for raising your score!**
> >
> > We sincerely appreciate your recognition of our work and your positive rating in support of its acceptance.

---

### Author Response · Authors · 2025-11-21
**(Part 1) General Response**

We sincerely thank the reviewers for their time and insightful feedback. We are encouraged by the recognition of EssenceBench’s novelty and are grateful for the constructive suggestions. To address these points and demonstrate the broad applicability of our framework in the era of frontier LLMs, we have conducted extensive new experiments and analyses during the rebuttal period.

**1. Expansion to Frontier, Agentic, Multimodal, and other Benchmarks**

To validate EssenceBench on modern and complex capabilities, we significantly expanded our evaluation scope beyond the original datasets. We applied our framework to **8 new challenging benchmarks** covering **Agentic Tool Use (LiveMCPBench)**, **Multimodal Reasoning (MathVista)**, and **Advanced Reasoning (GPQA, MATH, BBH, MUSR)**, as well as robustness and instruction following tasks. Across these diverse domains, EssenceBench consistently achieved State-of-the-Art compression performance. These results confirm that our method effectively captures the core signal of complex tasks—including those involving visual inputs and multi-step execution—proving its relevance for evaluating next-generation models. **We have provided results Table 16-23 in Appendix L in the revised manuscript.**

**2. Rigorous baseline comparison**

Following the reviewers' suggestions, we strengthened our comparative analysis by benchmarking against the SMART filtering method. Experiments on ARC and MMLU demonstrate that EssenceBench achieves comparable or superior ranking fidelity while offering significantly higher compression rates (up to 10x reduction). **We have provided results Table 24-25 in Appendix M in the revised manuscript.**

**3. Generalizability in unseen models**

Furthermore, we rigorously tested the generalizability of our compressed subsets on a hold-out set of **600 unseen models**. The strong correlation and low reconstruction error on these unseen models confirm that EssenceBench learns fundamental difficulty patterns rather than overfitting to the specific models used during compression. **We have provided results in Table 14 in Appendix J in the revised manuscript.**

**4. Ablation study on hyper-parameters and sensitivity evaluation**

We addressed concerns regarding hyperparameter sensitivity through comprehensive ablation studies. Our analyses of coarse filtering thresholds and Genetic Algorithm parameters reveal minimal performance variance, demonstrating that the framework is stable and not brittle to parameter tuning. **We have provided results Table 7-8 in Appendix F, Table 9-10 in Appendix G in the revised manuscript.**

**5. Intrisic properties of compressed datasets**

Additionally, we investigated the intrinsic properties of the compressed datasets, confirming that EssenceBench preserves the original data diversity and subject distribution with negligible selection bias. **We have provided results in Table 26-27 in Appendix N in the revised manuscript.**


**6. Efficiency analysis**

A detailed cost-benefit analysis further highlights the method's practicality, showing that the one-time compression cost is often lower than a single full-benchmark evaluation. **We have provided results in Table 15 in Appendix K in the revised manuscript.**

**7. Ablation study on ordering robustness in duplication filtering**

We addressed concerns regarding duplication order through comprehensive ablation studies. Our experiments of different orders reveal minimal performance variance, demonstrating that the ordering is robust in our coarse filtering process. **We have provided results Table 11-12 in Appendix H in the revised manuscript.**

**8. The neccessity of Genetic Algorithm (GA)**

To rigorously validate our choice of GA for the benchmark compression task, we conducted an extensive comparative analysis against two strong alternative optimization approaches: **K-means Clustering** (a geometric approach selecting centroids from sample embeddings) and **MLP-based Selection** (a gradient-based approach learning sample importance weights). Experiments demonstrate that our GA-based method achieves significant superiority, with substantially lower RMSE compared to both alternative methods. **We have provided detailed results in Table 13 in Appendix I in the revised manuscript.**

---

### Author Response · Authors · 2025-11-21
**(Part 2) General Response**

We explicitly thank the Area Chair and all reviewers (Reviewers hqMF, ykSp, MmMV, and TShU) for their time and constructive feedback. We are encouraged by the reviewers' shared recognition of the necessity for efficient LLM evaluation and by their positive assessment of our proposed EssenceBench framework.

We especially appreciate the acknowledgment that the high cost of evaluation is an "**important and practical problem**" (Reviewer ykSp) and an "**increasingly important**" challenge (Reviewer MmMV). We share the view that addressing the "**redundancy of the widely used LLM benchmarks**" (Reviewer hqMF) is essential for sustainable AI development. As noted by Reviewer hqMF, our work demonstrates the "**potential for a far smaller validation set**," which is a critical contribution for the community.

Regarding the methodology, we value the consensus on the soundness and clarity of our design. Reviewers noted that casting benchmark compression as an optimization problem is "**crisp**" (Reviewer TShU) and represents an "**effective paradigm**" (Reviewer MmMV). Specifically, the formalization of our pipeline was highlighted as "**well-motivated**" (Reviewer hqMF) and a design that reviewers explicitly appreciated regarding the formalization (Reviewer ykSp). Furthermore, the three-stage coarse-to-fine process was praised for being "**easy to follow**" (Reviewer TShU).

This rigorous design enables substantial efficiency gains, with reviewers commending our "**thorough and consistent**" (Reviewer ykSp) experiments that demonstrate "**strong empirical performance**" (Reviewer MmMV) and preserve ranking fidelity across diverse datasets.

---

### Author Response · Authors · 2025-11-26
**Looking forward to your response**

We thank the reviewers for their insightful feedback and recognition of our work’s value. We have taken every comment seriously and provided a detailed point-by-point rebuttal.

In response to the suggestions, **we performed an extensive series of experiments—totaling 27 new experiments—summarized in 21 tables and 6 figures.** These additions provide a more robust verification of our method's performance and generalization capabilities. The manuscript has been updated to reflect these findings, **with the main text extended by 1 page and the Appendix by **10** pages to ensure complete transparency.** We welcome any further feedback.

---

### Author Response · Authors · 2025-12-01
**Summary of Contributions and Rebuttal**

Dear PCs, SACS, ACs,

We would like to express our sincere appreciation for the reviewers’ dedication. Here we provide a general summary of the reviews and outline the efforts we have made during the discussion phase.

### **Claim of Contribution**
In this paper, we introduce EssenceBench, a coarse-to-fine framework for benchmark compression in large language model (LLM) evaluation. Our approach addresses the escalating scale of benchmark suites, which introduce computational inefficiencies and redundancy without commensurate improvements in evaluation fidelity. By conducting a sample-level redundancy analysis across text and ranking dimensions, framing compression as an optimization problem for score reconstruction, and employing an iterative Genetic Algorithm (GA) that integrates fitness-based subset search with attribution-based sample pruning, we systematically reduce benchmark sizes while preserving model performance rankings and accuracy.

### **Summary of Reviews and Responses**
In the current review, we are glad reviewers appreciate our work from multiple perspectives. Reviewer hqMF highlights **the potential of our 25× lossless to 200× near-lossless compression** for creating compact validation/test sets. Reviewer ykSp praises **the formalization as an optimization problem** with GA plus attribution refinement. Reviewer MmMV commends **addressing the key challenge of benchmark cost** in LLM evaluation while maintaining ranking reliability. Reviewer TShU appreciates **the clear, implementable three-stage design**.

We now address the specific points raised by the reviewers. All concerns have been carefully addressed in the table below.


| Reviewer   | Reviewer's Concern/Questions                                 | Author's Response                                            |
| ---------- | ------------------------------------------------------------ | ------------------------------------------------------------ |
| hqMF, ykSp | More benchmark compression results                           | We expanded evaluation to 8 new benchmarks (e.g., LiveMCPBench for agentic tools, MathVista for multimodal, GPQA/MATH/BBH/MUSR for reasoning, IFEVAL for instruct following and GSM-Plus for math robustness, adversarial evaluation); results showed SOTA compression across domains, confirming capture of complex task signals. |
| ykSp       | Rigorous baseline comparison                                 | We benchmarked against SMART filtering on ARC and MMLU; results attained comparable/superior ranking fidelity with up to 10× higher compression. |
| ykSp       | Generalizability in unseen models                            | We tested on hold-out set of 600 unseen models; results showed strong correlation and low reconstruction error, proving learning of fundamental patterns without overfitting. |
| ykSp, MmMV | Ablation study on hyper-parameters and sensitivity evaluation | We performed ablations on coarse filtering thresholds and GA parameters; results revealed minimal performance variance, confirming stability and non-brittleness. |
| hqMF       | Intrinsic properties of compressed datasets                  | We investigated properties like diversity and subject distribution; results preserved originals with negligible bias. |
| ykSp, MmMV | Efficiency analysis                                          | We conducted cost-benefit analysis; results showed one-time compression cost often lower than single full evaluation. |
| TShU       | Ablation study on ordering robustness in duplication filtering | We ran ablations with different orders in coarse filtering; results found minimal variance, proving robustness. |
| ykSp       | The necessity of Genetic Algorithm (GA)                      | We compared GA to K-means and MLP-based alternatives; results demonstrated superior performance with lower RMSE. |


### **Reviewers' Rating Changes Pre- and Post-Rebuttal**
| Reviewer | Initial Rating/Confidence | Post-Rebuttal Rating/Confidence|
|---|---|---|
|hqMF|4/4|**6/4**|
|ykSp|6/4|**6/4**|
|MmMV|4/3|**6/3**|
|TShU|4/4|No response yet|

We sincerely hope to engage in constructive dialogue with reviewers as we believe this exchange is vital for improving the quality of our work.

Best regards,

Authors of submission 1578

---

### Meta-Review · Area_Chair_ohFi · 2026-01-06

**Summary:**

The paper proposes EssenceBench, a framework designed to compress LLM benchmarks to reduce evaluation costs while maintaining ranking fidelity. The method utilizes a three-stage pipeline: coarse filtering based on text and ranking redundancy, a Genetic Algorithm (GA) for subset selection optimizing for score reconstruction, and an attribution-based refinement step. The study includes comparisons of different compression baseline methods, and ablation studies on each stage of the method. During the rebuttal period, the authors conducted additional work to address reviewer concerns. The rebuttal strengthened the paper and addressed the reservations regarding generalizability and baseline comparisons.

Reviewers initially give 4/4/4/6, but most of reviewers raise their score to 6 after rebuttal. However, they also provided few suggestions for improvement, which the authors should refine in the next version.

**Reviewer Concerns:**

1.  **Benchmark Scope and Validity (Reviewer hqMF):** The reviewer was concerned that the original experiments relied on "saturated" benchmarks. The authors addressed this by extending the evaluation to 8 new benchmarks including agentic (LiveMCPBench), multimodal (MathVista), and reasoning (GPQA, MATH) tasks.

2.   **Baselines and Comparisons (Reviewer ykSp):** The request for a comparison against SMART filtering was addressed. Results showed EssenceBench achieves comparable or superior fidelity with better compression performance. Also, the authors incorporated SMART filtering as an alternative of the stage 1.

3.   **Generalizability (Reviewer ykSp):** Concerns about overfitting to the training models were mitigated by a new hold-out experiment by the authors, showing high correlation (>0.99) and low reconstruction error.

4.   **Hyperparameter Sensitivity (Reviewers ykSp, MmMV, TShU):** The authors provided sensitivity analyses for filtering thresholds ($\tau_{text}$, $\tau_{ranking}$), GA population size, and elite size, demonstrating that the method is robust to parameter variations.

5.  **Ordering Bias (Reviewer TShU):** The concern that the retention strategy introduced bias was addressed via an ablation study comparing sequential, random, and conservative filtering.

6.  **Intrinsic Properties (Reviewer MmMV):** The authors analyzed the diversity and subject distribution of the compressed sets, confirming that the method introduces negligible selection bias compared to the full sets.

7.  **Computational Cost (Reviewer ykSp, MmMV):** A cost-benefit analysis was provided, showing that the one-time compression cost is often lower than a single full-benchmark evaluation, with a break-even point of less than one evaluation cycle.

**Reviewer Scores:**

*   **Reviewer hqMF:** **6** (Raised from 4 after the inclusion of new benchmarks, claimed by authors).
*   **Reviewer ykSp:** **6** (Maintained score; confirmed concerns were addressed).
*   **Reviewer MmMV:** **6** (Raised from 4 after additional experiments, claimed by authors).
*   **Reviewer TShU:** **4** (The reviewer did not post a final acknowledgment, but their primary technical concern regarding ordering bias was empirically addressed by the ablation study in the rebuttal. Given their initial rating of 4, it is possible to raise the scores).

---

### Decision · Program_Chairs · 2026-01-26

Accept (Poster)